# Silicate dissolution boosts the $CO_2$ concentrations in subduction fluids

S. Tumiati [1], C. Tiraboschi[2], D.A. Sverjensky[3], T. Pettke[4], S. Recchia[5], P. Ulmer[6], F. Miozzi[1,7] & S. Poli [1]

Estimates of dissolved $CO_2$ in subduction-zone fluids are based on thermodynamic models, relying on a very sparse experimental data base. Here, we present experimental data at 1–3 GPa, 800 °C, and $\Delta FMQ \approx -0.5$ for the volatiles and solute contents of graphite-saturated fluids in the systems COH, $SiO_2$–COH ( + quartz/coesite) and $MgO$–$SiO_2$-COH ( + forsterite and enstatite). The $CO_2$ content of fluids interacting with silicates exceeds the amounts measured in the pure COH system by up to 30 mol%, as a consequence of a decrease in water activity probably associated with the formation of organic complexes containing Si–O–C and Si–O–Mg bonds. The interaction of deep aqueous fluids with silicates is a novel mechanism for controlling the composition of subduction COH fluids, promoting the deep $CO_2$ transfer from the slab–mantle interface to the overlying mantle wedge, in particular where fluids are stable over melts.

[1] Dipartimento di Scienze della Terra, Università degli Studi di Milano, via Mangiagalli 34, 20133 Milano, Italy. [2] Dipartimento di Scienze dell'Ambiente e della Terra, Università degli Studi di Milano Bicocca, Piazza della Scienza 4, 20126 Milano, Italy. [3] Department of Earth & Planetary Sciences, Johns Hopkins University, Baltimore, MD 21218 USA. [4] Institute of Geological Sciences, University of Bern, Baltzerstrasse 1+3, 3012 Bern, Switzerland. [5] Dipartimento di Scienza e Alta Tecnologia, Università degli Studi dell'Insubria, via Valleggio 11, 22100 Como, Italy. [6] Institute of Geochemistry and Petrology, ETH Zürich, Clausiusstrasse 25 / NW E77, 8092 Zürich, Switzerland. [7] Institut de Minéralogie, de Physique des Matériaux, et de Cosmochimie (IMPMC), Sorbonne Universités – UPMC, UMR CNRS, 7590, Muséum National d'Histoire Naturelle, IRD UMR 206, 75005 Paris, France. Correspondence and requests for materials should be addressed to S.T. (email: simone.tumiati@unimi.it)

Subduction of the oceanic lithosphere and its sedimentary cover is accompanied by devolatilization processes[1]. $CO_2$ removal through dissolution of carbonates occurring in altered oceanic lithosphere and its sedimentary cover, along with diapirism of slab rocks[2, 3] and/or melts[4], provides an efficient way to recycle carbon back to the mantle wedge and, ultimately, to the Earth's surface[5, 6]. However, other forms of carbon, often closely associated with silicates, have been reported in slab rocks and in particular in subduction mélanges. For instance, graphite has been described in blueschist-facies mafic rocks, metasediments and hybridized peridotites at Santa Catalina[7], and in ophiolitic serpentinites from the Western Alps[8], where also diamond has been found in UHP metasediments[9]. In Alpine Corsica[10] and in the Western Italian Alps[11], reduction of carbonates during subduction results in graphite-rich metasediments and serpentinites, suggesting that graphite may become a major phase in hybridized silicate-rich subduction mélanges. Graphite has been considered to represent a refractory sink of carbon in the subducting slab, owing to its lower solubility in aqueous fluids[12] and melts[13] compared to carbonates. On the other hand, graphite dissolution mechanisms and solute transport in complex COH fluids at high pressures have remained experimentally unconstrained. Moreover, recent thermodynamic models highlight the role of graphite in subduction-zone fluids[14] and suggest that the presence of graphite is capable of modifying fluid properties and promoting the formation of C-bearing anions, possibly enhancing the complexation of major and trace elements at elevated $P$ and $T$ conditions[12].

Here, we provide comprehensive experimental constraints on the composition of high-pressure graphite-saturated COH fluids in terms of dissolved $CO_2$, $SiO_2$, and MgO in increasingly complex petrological systems at controlled redox conditions, buffered by using the double-capsule technique and both the nickel-nickel oxide (NNO) and the fayalite-magnetite-quartz (FMQ) buffers, in order to develop a model for the interaction between deep aqueous fluids and silicates in subduction mélanges. A carbonate-free compositional range has been explored at $P = 1$ GPa, $T = 800\,°C$ and $P = 3$ GPa, $T = 800\,°C$ in order to focus on the role of graphite and silicates in the investigated processes. We synthesized COH fluids in equilibrium with graphite and other minerals representative of subduction mélanges, i.e., Mg-silicates (forsterite and enstatite), representative of the mantle component, and quartz, representative of the sedimentary component. Experimental products were analyzed for their volatile COH composition by quadrupole mass spectrometry (QMS) and for their Mg and Si solute load by cryogenic laser-ablation inductively coupled plasma mass spectrometry (ICP-MS). Measured data were compared to thermodynamic modeling results. Further details are provided in Methods section and as Supplementary Information. Our results suggest that the interaction of deep aqueous fluids with silicates in the presence of graphite in a subduction mélange promotes the dissolution of graphite and enhances the $CO_2$ contents of the fluids; this provides a new mechanism for controlling the volatile composition of COH fluids already at depths of ~30 km.

## Results

**$CO_2$ contents of fluids in equilibrium with silicates**. The volatile compositions of COH fluids were measured by piercing the capsules after quench in a gas-tight vessel and then conveying the emanating gases to a quadrupole mass spectrometer (QMS)[15]. Measured data were subsequently compared with the compositions predicted by traditional thermodynamic modeling using different equations of state and mixing properties of $H_2O$ and non-polar species (details in Methods section). Carbon-saturated

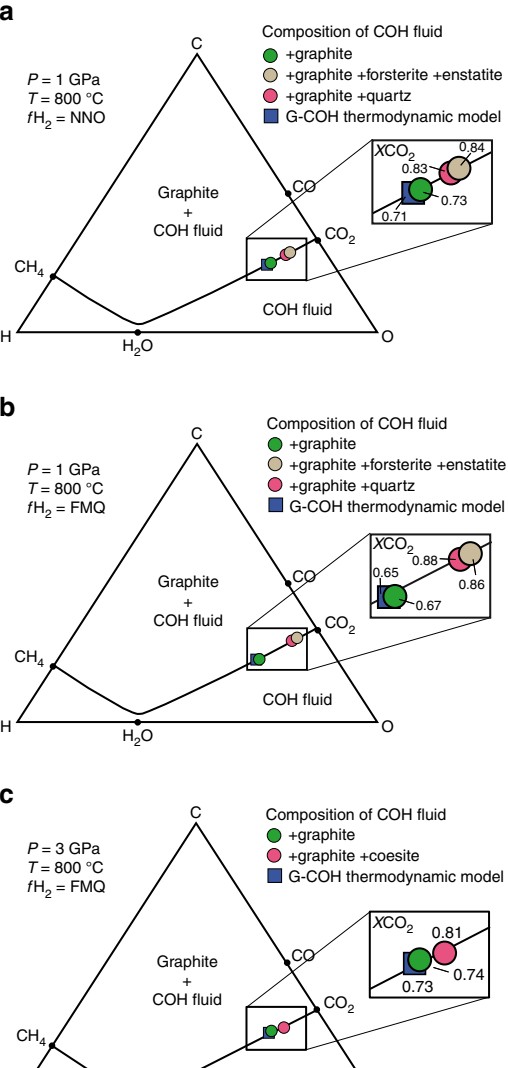

**Fig. 1** Measured volatile composition of graphite-saturated fluids interacting with silicates. C–O–H diagram showing the volatile composition of fluids, measured by quadrupole mass spectrometry, synthesized at $P = 1$ GPa, $T = 800\,°C$ and $f{H_2}^{NNO}$ **a** and $f{H_2}^{FMQ}$ **b** conditions, and $P = 3$ GPa, $T = 800\,°C$ and $f{H_2}^{FMQ}$ **c** in equilibrium with graphite only (COH system; *green dots*), graphite + forsterite + enstatite (MgO–SiO$_2$–COH system; *ochre dots*) and graphite + quartz (SiO$_2$–COH system; *pink dots*). The graphite-saturation surfaces, i.e., the loci of points representing all possible compositions of graphite-saturated COH fluids (G-COH) at fixed $P$, $T$ and variable $fO_2$ conditions, calculated by thermodynamic modeling[45], are shown for comparison (*black curves*) along with the volatile fluid composition predicted by the EoS of Zhang and Duan[46] using the $H_2$ fugacity coefficient of Connolly and Cesare[45] (*blue squares*; Methods, Supplementary Fig. 2 and Supplementary Table 2 for details). Compared to the pure COH system, the $XCO_2$ [ $= CO_2/(H_2O + CO_2)$] of the fluids, shown in the zoom boxes, increases in silicate-bearing systems

fluids were synthesized first in the pure COH system, where fluids interacted only with graphite (Supplementary Fig. 1a). As predicted by thermodynamic modeling, all analyzed fluids are located on the graphite-saturation surfaces (*black lines* in Fig. 1). The measured $XCO_2$ [ $= CO_2/(H_2O + CO_2)_{molar}$] (Supplementary Table 1) and modeled compositions (Supplementary Table 2) of the COH fluids coexisting only with graphite overlap in both

experiments buffered with NNO and FMQ (Fig. 1 and Supplementary Fig. 2). However, experiments where COH fluids coexist with either quartz/coesite (SiO$_2$–COH system; Supplementary Fig. 1c) or forsterite + enstatite in addition to graphite (MgO–SiO$_2$–COH system; Supplementary Fig. 1b) clearly display a significant increase of $X$CO$_2$ (Fig. 1; Supplementary Table 1). The increase in $X$CO$_2$ is + 33% (MgO–SiO$_2$–COH) and + 28% (SiO$_2$–COH) in experiments buffered by FMQ at 1 GPa, 800 °C, + 9% (SiO$_2$–COH) in experiments buffered by FMQ at 3 GPa, 800 °C, and + 14% (SiO$_2$–COH) and + 15% (MgO–SiO$_2$–COH) in experiments buffered by NNO at 1 GPa, 800 °C.

**Si and Mg contents of fluids in equilibrium with silicates**. The interpretation of the observed differences in volatile content of COH fluids that do and do not interact with silicates is not straightforward. Because, we did not find any evidence of hydration and/or carbonation reactions in the run products, our preferred interpretation is that dissolution reactions of silicates are influencing the $X$CO$_2$ of the fluid, provided that graphite is present in excess. We therefore measured additionally the dissolved Si and Mg in the synthetic COH fluids at 1 GPa and 800 °C by using a modified version of the cryogenic LA-ICP-MS technique[16]. This technique was originally developed for the analysis of solutes in pure water, which is frozen in the experimental capsules and analyzed via laser-ablation ICP-MS. In the case of COH fluids, an immiscible mixture of water and non-polar gases is expected at the investigated $P$, $T$ conditions[17] (inset in Supplementary Fig. 1e). The cryogenic technique, operating at approximately −35 °C, only keeps water in a solid state. Therefore, the other volatiles (including CO$_2$) are lost when the capsule is opened for analysis. Consequently, the solute contents retrieved by ICP-MS pertain to the aqueous part of the bulk COH fluid only (Supplementary Table 3; other details in Methods section). In Fig. 2, we report the solubility data of quartz (SiO$_2$–COH system) and forsterite + enstatite (MgO–SiO$_2$–COH system) at 1 GPa and 800 °C, where SiO$_2$ molalities have been obtained by correcting the concentration of the internal standard (Cs) on the basis of the measured fluid $X$CO$_2$ (Supplementary Fig. 3; Supplementary Table 3). The measured dissolved Si in COH fluids in equilibrium with quartz and graphite at 1 GPa, 800 °C and $f$H$_2^{NNO}$ conditions ($X$CO$_2$ = 0.83; Supplementary Table 1; Supplementary Fig. 1d) is 0.30 ± 0.04 mol kgH$_2$O$^{-1}$, which is much lower than the quartz solubility in pure water (1.23 mol kgH$_2$O$^{-1}$ ref. [18]), but much higher than previously reported quartz solubilities in H$_2$O–CO$_2$ fluids characterized by similar $X$CO$_2$ but without graphite (0.04 mol kgH$_2$O$^{-1}$ ref. [19]. for $X$CO$_2$ = 0.75; 0.01 mol kgH$_2$O$^{-1}$ for $X$CO$_2$ = 0.94 ref. [20]). The dissolved silica in COH fluids in equilibrium with forsterite, enstatite and graphite ($X$CO$_2$ = 0.84; Supplementary Table 1; Supplementary Fig. 1e) is much higher compared to the SiO$_2$–COH system (1.24 ± 0.19 mol kgH$_2$O$^{-1}$), resembling the solubility of quartz in pure water. The solubility of forsterite and enstatite has not previously been measured in mixed H$_2$O–CO$_2$ fluids, but it has been investigated in pure water[21–24], amounting to 0.21–0.30 mol SiO$_2$ kgH$_2$O$^{-1}$ ref. [24]. We additionally performed a dissolution experiment of forsterite and enstatite in pure water (MgO–SiO$_2$–H$_2$O system; *black dot* in Fig. 2), obtaining very similar results (0.22 ± 0.06 mol SiO$_2$ kgH$_2$O$^{-1}$), clearly testifying that the solubility of forsterite and enstatite in pure water is much lower than in COH-bearing fluids.

**Constraints on the increased solubility of silicates**. Our solubility results indicate that in both the graphite-saturated SiO$_2$–COH and MgO–SiO$_2$–COH systems, the carbon dissolved in fluids does not behave merely as an inert diluent, but promotes

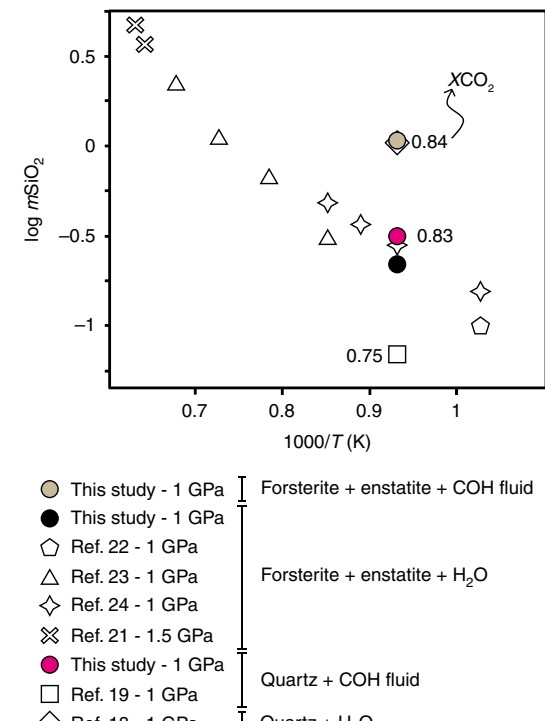

**Fig. 2** Measured dissolved SiO$_2$ in graphite-saturated fluids interacting with silicates. Solute contents in experimental fluids at $P$ = 1 GPa, $T$ = 800 °C and $f$H$_2^{NNO}$ in equilibrium with graphite + forsterite + enstatite (MgO–SiO$_2$–COH system; *ochre dot*), and with graphite + quartz (SiO$_2$–COH system; *pink dot*), expressed as moles of SiO$_2$ per kg of water, measured by cryogenic laser-ablation ICP-MS. Solute content of forsterite + enstatite in pure water (*black dot*) and selected literature solubility data in the systems SiO$_2$–H$_2$O, SiO$_2$–COH and MgO–SiO$_2$–H$_2$O are shown for comparison. The measured dissolved silica is 0.22 mol kgH$_2$O$^{-1}$ in the MgO–SiO$_2$–H$_2$O system, 0.30 mol kgH$_2$O$^{-1}$ in the SiO$_2$–COH system ($X$CO$_2$ = 0.83) and 1.24 mol kgH$_2$O$^{-1}$ in the MgO–SiO$_2$–COH system ($X$CO$_2$ = 0.84)

the dissolution of silicates at the conditions of our experiments. The simplest explanation of these results is that new unexpected organic complexes containing Si–O–C and Si–O–Mg bonds are formed in addition to the solutes known experimentally, such as the Mg$^{2+}$ ion and the silica monomer and dimer[24, 25]. This hypothesis is supported by the measured dissolved Mg concentrations in the MgO–SiO$_2$–H$_2$O and MgO–SiO$_2$–COH systems. Although published experimental data are not available to date for the $P$–$T$ conditions investigated, the extrapolated solubility of Mg derived from the dissolution of forsterite and enstatite in pure water should be less than 0.17 mol kgH$_2$O$^{-1}$ at $P$ = 1–2 GPa and $T$ = 900–1200 °C[23]. In our experiments in pure water at 1 GPa and 800 °C, dissolved Mg is slightly higher (0.28 ± 0.04 mol kgH$_2$O$^{-1}$; Supplementary Table 3). However, in COH fluids at identical $P$, $T$ conditions the Mg content almost quadruples (1.08 ± 0.10 mol kgH$_2$O$^{-1}$; Supplementary Table 3).

In order to gain some insight into the possible aqueous species relevant to the different types of experimental systems reported in the present study, we used a thermodynamic model that also takes Mg- and Si-bearing dissolved species into account, in addition to neutral COH species, and therefore is suitable for the systems SiO$_2$–COH and MgO–SiO$_2$–COH. We first performed preliminary calculations using the aqueous speciation-solubility code EQ3[26] adapted to include equilibrium constants calculated with the Deep Earth Water model[27, 28] at 1 GPa and 800 °C (other details in Methods section). For the system

MgO–SiO$_2$–H$_2$O, the predicted silica concentrations agreed with the experimentally measured values. However, the predicted Mg concentrations in the MgO–SiO$_2$–H$_2$O system were too low, even when a predicted equilibrium constant for the species Mg(OH)$^+$ was included. Consequently, an Mg(OH)$_{2(aq)}$ complex was fit to the experimental solubility data for the MgO–SiO$_2$–H$_2$O system (Supplementary Table 4).

In the system MgO–SiO$_2$–COH, the new Mg(OH)$_{2(aq)}$ complex and the predicted complexes MgHCO$_3$$^+$, MgCO$_{3(aq)}$, MgHSiO$_3$$^+$, and MgSiO$_3$(aq) were used together with the silica monomer and dimer in trial calculations to predict Mg and Si solubilities. However, numerous calculations resulted in solubilities that were too low, indicating the likely need for an additional complex involving the components MgO, CO$_2$, and SiO$_2$ in the MgO–SiO$_2$–COH system compared to the carbon-free MgO–SiO$_2$–H$_2$O system. An analysis of the available solubility data at 800 °C and 1 GPa for the MgO–SiO$_2$–H$_2$O and the MgO–SiO$_2$–COH systems indicates that a variety of MgO–SiO$_2$–CO$_2$ complexes might be feasible. For example, complexes involving oxidized C-species such as bicarbonate or carbonate complexes, or complexes involving reduced C-species could account for the observed solubilities. In Supplementary Table 4, we present the results for a complex of Mg with the silicate anion and the organic anion propionate (MgSiC complex) which can be written as Mg[OSi(OH)$_3$][CH$_3$CH$_2$COO]. This complex provides an explanation for the distinctive enhanced solubilities of Mg and Si measured in the present study. In particular, the MgSiC complex is predicted to be important in the relatively reduced systems investigated, i.e., bearing H$_2$O–CO$_2$ fluids in equilibrium with graphite close to the C–CO$_2$ (CCO) buffer[29]. This complex, however, would be insignificant in the COH fluids in all previous studies of silicate solubilities that focused on unbuffered H$_2$O–CO$_2$ fluids without graphite (i.e., above the CCO buffer), which are potentially stable up to extremely oxidizing conditions. Analogous calculations for the SiO$_2$–COH system suggest the possibility of other organic complexes involving SiO$_2$ and reduced C-species. It is worth noting here that the stability of organic COH species in subduction-zone fluids has been recently suggested[14]. A more complete analysis of the potential importance of MgO–SiO$_2$–CO$_2$ complexes over a wide range of temperatures and pressures is hampered by the lack of other experimental Mg- and Si-solubility data in the MgO–SiO$_2$–COH and SiO$_2$–COH systems. Consequently, a full equation of state characterization of the standard partial molal properties of MgO–SiO$_2$–CO$_2$ complexes must await the development of estimation schemes for refining the properties of such complexes.

**Constraints on the increase of CO$_2$.** Analytical results indicate that in both the SiO$_2$–COH and MgO–SiO$_2$–COH systems externally buffered by FMQ and NNO, dissolution reactions of either quartz/coesite and forsterite + enstatite are able to boost the CO$_2$ content of graphite-saturated COH fluids at elevated pressures and temperatures. No significant differences have been observed concerning the XCO$_2$ increase between quartz-bearing and forsterite + enstatite-bearing experiments, suggesting that dissolved Si, rather than Mg, is the major player in this boosting process. This could be attributed to the polymerized nature of silica at high pressures and temperatures[20], which may be different from other species, for example aluminum species[30]. Our model predicts that neutral silica monomers [SiO$_2$(aq)] and dimers [Si$_2$O$_{4(aq)}$] are important species at the investigated conditions (Supplementary Table 4), which is supported also by experimental data from forsterite and enstatite incongruent dissolution in pure H$_2$O at 1 GPa and 700 °C[24]. However, none of

the models we developed was able to account for the large increase in the CO$_2$ solubility in the fluid that was experimentally determined in SiO$_2$–COH or MgO–SiO$_2$–COH systems. Several possible explanations for this novel effect can be suggested. For instance, very large amounts of HCO$_3$$^-$ or CO$_3$$^{2-}$ species that would occur in the SiO$_2$-bearing fluids could be converted to CO$_2$ through reactions of the type:

$$CO_3^{2-} + 2H^+ \rightleftharpoons CO_2 + H_2O. \tag{1}$$

However, theoretical model results indicate that at 1 GPa and 800 °C insignificant amounts of HCO$_3$$^-$ or CO$_3$$^{2-}$ are present in the fluids. Actually, the association forsterite + enstatite at 0.5 GPa and 600 °C buffers the pH of the coexisting aqueous fluid to between 2.5 and 4 ref. [14]. Our model predicts that at 1 GPa and 800 °C the pH of the fluid is 5.57 in the MgO–SiO$_2$–H$_2$O system and 3.73 in the MgO–SiO$_2$–COH system (Supplementary Table 4). These pH values would favor the stability of the species CO$_{2(aq)}$ against HCO$_3$$^-$ and CO$_3$$^{2-}$. Moreover, reaction (Eq. 1) is independent on the presence of silica, thus it cannot explain the influence exerted by silicate dissolution in enhancing the XCO$_2$ of the fluid.

Alternatively, a change in fCO$_2$ at fixed fH$_2$ imposed by the buffers could result from a change in fH$_2$O or fO$_2$ associated with dissolved silica. In our double-capsule experiments, silicate dissolution reactions in both the MgO–SiO$_2$–COH and SiO$_2$–COH systems proceed together with the dissolution of graphite. In the pure COH system, the dissolution of graphite at the investigated relatively oxidizing conditions is controlled by the reaction C + 2H$_2$+2O$_2$ $\rightleftharpoons$ CO$_2$+2H$_2$O (see Eq. 12 in Methods section; Supplementary Fig. 4) until the fugacity of H$_2$ in the inner capsule, containing a COH fluid, equals that in the outer capsule, containing C-free water in equilibrium with the NNO or the FMQ buffers. The equilibrium constant of the reaction above is:

$$K = \frac{fCO_2 \times (fH_2O)^2}{(fH_2)^2 \times (fO_2)^2}. \tag{2}$$

Our experimental data (runs COH70 and COH69 in Supplementary Table 1) and the thermodynamic model of Zhang and Duan (ZD09mod in Supplementary Table 2; details in Methods section) allow retrieving K in the pure COH system at 1 GPa and 800 °C resulting both in the NNO- and the FMQ-buffered experiments a value of log K = 37.5, assuming log fH$_2$$^{NNO}$ = 1.775 and log fH$_2$$^{FMQ}$ = 1.889, and log fO$_2$ = −14.28 (inner capsule buffered by NNO) and log fO$_2$ = −14.31 (inner capsule buffered by FMQ), respectively.

The observed increase in fluid XCO$_2$ in SiO$_2$–COH and MgO–SiO$_2$–COH systems will result in an increase of fCO$_2$. Therefore, it is convenient to express Eq. 2 as a function of fCO$_2$:

$$fCO_2 = \frac{K \times (fH_2)^2 \times (fO_2)^2}{(fH_2O)^2}, \tag{3}$$

and

$$\log fCO_2 = \log K + 2\log fH_2 + 2\log fO_2 - 2\log fH_2O. \tag{4}$$

By fixing log fH$_2$ and log K, fCO$_2$ is expressed as a function of the two variables fO$_2$ and fH$_2$O, which can be represented as two tri-dimensional surfaces, one calculated for log fH$_2$$^{NNO}$ (Fig. 3a) and one calculated for log fH$_2$$^{FMQ}$ (Fig. 3b). In order to move from the fCO$_2$ of the pure COH system (*green dots* in Fig. 3) to

**a**

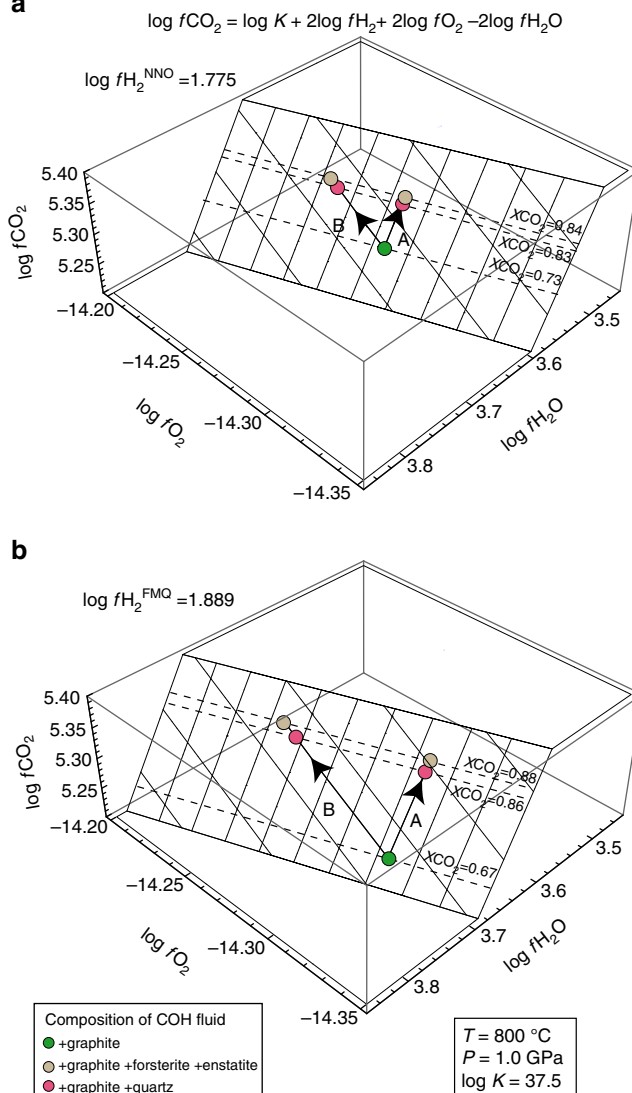

$$\log f\mathrm{CO_2} = \log K + 2\log f\mathrm{H_2} + 2\log f\mathrm{O_2} - 2\log f\mathrm{H_2O}$$

$\log f\mathrm{H_2}^{\mathrm{NNO}} = 1.775$

**b**

$\log f\mathrm{H_2}^{\mathrm{FMQ}} = 1.889$

Composition of COH fluid
- 🟢 +graphite
- 🟤 +graphite +forsterite +enstatite
- 🔴 +graphite +quartz

$T = 800\ ^\circ\mathrm{C}$
$P = 1.0\ \mathrm{GPa}$
$\log K = 37.5$

**Fig. 3** Model for increased dissolved $CO_2$ in graphite-saturated fluids interacting with silicates. 3D plot of the equations $\log f\mathrm{CO_2} = \log K + 2\log f\mathrm{H_2}^{\mathrm{NNO}} + 2\log f\mathrm{O_2} - 2\log f\mathrm{H_2O}$ **a** and $\log f\mathrm{CO_2} = \log K + 2\log f\mathrm{H_2}^{\mathrm{FMQ}} + 2\log f\mathrm{O_2} - 2\log f\mathrm{H_2O}$ **b**, assuming $\log K = 37.5$, retrieved from the experimental data and thermodynamic modeling at 1 GPa and 800 °C, using the EoS of Zhang and Duan[46] and the $H_2$ fugacity coefficient of Connolly and Cesare[45]. The increase in $XCO_2$ observed experimentally in the systems $SiO_2$–COH (*pink dots*) and MgO–$SiO_2$–COH (*ochre dots*), compared to the pure COH system (*green dots*), can be achieved either by decreasing $f\mathrm{H_2O}$ (*arrays A*; preferred interpretation) or by increasing $f\mathrm{O_2}$ (*arrays B*; unlikely because of the lack of Fe and other redox sensitive elements in the considered silicates). The decrease in $f\mathrm{H_2O}$ is ascribed to dissolved Si-complexes and almost independent on dissolved Mg-complexes, and therefore attributable to formation of hydrated silica monomers [$Si(OH)_4$] and dimers [$Si_2O(OH)_6$]

the $f\mathrm{CO_2}$ retrieved from measurement in the $SiO_2$–COH (*pink dots*) and MgO–$SiO_2$–COH (*ochre dots*), either $\log f\mathrm{H_2O}$ (i.e., $H_2O$ activity) should decrease at constant $\log f\mathrm{O_2}$ (arrays A in Fig. 3) or $\log f\mathrm{O_2}$ should increase at constant $\log f\mathrm{H_2O}$ (arrays B in Fig. 3). This model predicts that very small variations in either $\log f\mathrm{H_2O}$ or $f\mathrm{O_2}$ (~0.03 log units at $f\mathrm{H_2}^{\mathrm{NNO}}$; ~0.06 log units at $f\mathrm{H_2}^{\mathrm{FMQ}}$) can account for the measured increase in $f\mathrm{CO_2}$ in both $SiO_2$–COH and MgO–$SiO_2$–COH systems (Supplementary Table 5). However, in view of the absence of redox sensitive

components in the minerals under investigation, even small variations of $f\mathrm{O_2}$ are highly unlikely in our experimental system and therefore we suggest that a decrease in $f\mathrm{H_2O}$ is the culprit of the observed increase in $f\mathrm{CO_2}$ in the MgO–$SiO_2$–COH and $SiO_2$–COH systems. By using the $H_2O$ fugacity coefficient from Zhang and Duan[31] (1.579 at 1 GPa and 800 °C), we are able to calculate the decrease in water activity ($a\mathrm{H_2O}$), assuming that $a\mathrm{H_2O} = f\mathrm{H_2O}/f\mathrm{H_2O}^0$, where $f\mathrm{H_2O}^0$ is the fugacity of pure water at 1 GPa and 800 °C (i.e., 15785.5). The required decrease in $a\mathrm{H_2O}$ with respect to the pure COH system is −4.17 log units in the experiments buffered with NNO and −4.14 log units in the experiments buffered with FMQ. Because only small differences in measured $XCO_2$ have been observed between the $SiO_2$–COH and MgO–$SiO_2$–COH systems, we argue that dissolved silica monomers [$Si(OH)_4$] and dimers [$Si_2O(OH)_6$] are much more effective than MgO–$SiO_2$–$CO_2$ complexes in decreasing water activity. The activity of total silica can be calculated on the basis of the measured $SiO_2$ molality in the experiments buffered with NNO and the measured $XCO_2$ of the corresponding COH fluid[20], $\log(a\mathrm{SiO_2})$ being equal to −2.59 in MgO–$SiO_2$–COH and −2.90 in the $SiO_2$–COH systems. The activity of dissolved silica in the $SiO_2$–COH and MgO–$SiO_2$–COH systems is therefore much higher (about 20 times in $SiO_2$–COH system; 40 times in MgO–$SiO_2$–COH system) compared to the difference in $a\mathrm{H_2O}$ estimated in the $SiO_2$–COH and MgO–$SiO_2$–COH systems versus the pure COH system. In order to match the observed increase in $f\mathrm{CO_2}$, we estimated that only 0.31 mol% ($SiO_2$–COH system; i.e., 0.006 $m\mathrm{SiO_2}/\mathrm{kgH_2O}$) and 1.89 mol% (MgO–$SiO_2$–COH system; i.e., 0.004 $m\mathrm{SiO_2}/\mathrm{kgH_2O}$) of the measured dissolved silica are required, assuming that the decrease in water activity is solely related to the formation of hydrated silica monomers [$Si(OH)_4$] and dimers [$Si_2O(OH)_6$]. These low solubility data (cf. also Supplementary Table 4) are almost identical to quartz solubility in graphite-free systems bearing very high-$XCO_2$ $H_2O$–$CO_2$ fluids[20], strongly supporting the hypothesis that additional new $SiO_2$–$CO_2$ and MgO–$SiO_2$–$CO_2$ complexes are required to account for the surprisingly high total dissolved silica measured in our experiments.

**Dissolution of graphite and silicates in subduction mélanges**. Our results suggest that the silica component derived from the dissolution of either magnesium silicates or quartz/coesite alone, even in absence of carbonates, controls the composition of deep COH fluids in equilibrium with graphite, in particular enhancing their $CO_2$ content when compared to $SiO_2$-free systems. This mechanism could be effective especially in cold subduction zones, where subsolidus conditions prevail, and particularly in subduction mélanges, where silicate minerals and graphite[10] are thought to be abundant and flushed by aqueous fluids originating from the dehydration of the subducted lithosphere[32, 33] (Fig. 4). Independently from the occurrence of carbonates, the dissolution of silicates can boost the dissolution of graphite in the subduction mélange in the form of volatile $CO_2$ dissolved in COH fluids by up to + 30% compared to silicate-free systems. These $CO_2$-rich fluids will interact with the overlying mantle rocks, influencing metasomatic processes, carbonation/decarbonation reactions, and the melting temperatures of rocks in the mantle wedge[3]. From this perspective, the fact that fluid inclusions in shallow-mantle xenoliths are often dominated by $CO_2$ over water[34] could be an effect of the inherited $CO_2$-rich composition of slab-derived fluids, and does not necessarily require extensive diffusional hydrogen loss from the inclusion to the host mineral. Moreover, as this $CO_2$ boosting effect cannot be predicted by available thermodynamic models that have been used to estimate the amount of $CO_2$ recycled from subducted carbon-bearing

**a**

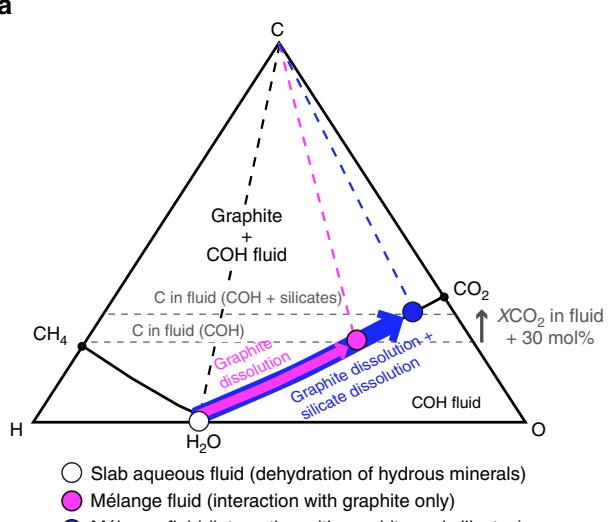

○ Slab aqueous fluid (dehydration of hydrous minerals)
● Mélange fluid (interaction with graphite only)
● Mélange fluid (interaction with graphite and silicates)

**b**

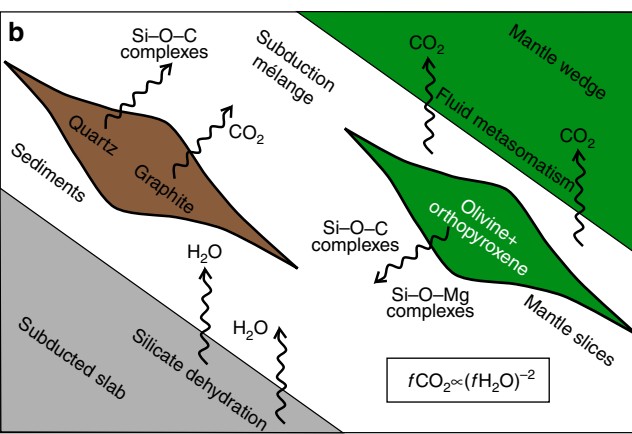

**Fig. 4** Evolution of dissolved $CO_2$ in fluids flushing a subduction mélange. **a** C–O–H diagram showing the compositional evolution of an aqueous fluid (*empty dot*) from a slab interacting with graphite alone (*magenta dot*) and graphite + silicates (*blue dot*) in the subduction mélange. Compared to fluids interacting with graphite alone, fluids in equilibrium with graphite + silicates display markedly higher $XCO_2$ (up to + 30 mol% at 1 GPa, 800 °C and $fH_2^{FMQ}$); **b** schematic drawing of a subduction mélange typical of collisional orogens flushed by aqueous fluids released from the underlying dehydrating slab. In the model, these fluids interact with slices of rocks bearing forsterite + enstatite (mantle-derived) and graphite + quartz (sedimentary slab cover). Because of the quadratic growth, small changes in water fugacity/activity due to dissolution of silicates and the consequent release of organosilicon complexes, can enhance the $CO_2$ content in COH fluids, even in the absence of carbonates, as long as graphite is present. Such fluids will flush the overlying mantle wedge, influencing metasomatic and/or melting processes

sediments[5], the estimated carbon transfer linked to the oxidation of sedimentary organic carbon and graphite (~6 Mt C/y[5]) probably needs to be adjusted up by 10–30% (0.6–1.8 Mt C/y). Additional experiments investigating more complex, carbonate-bearing systems as well as an improved quantification of the content of graphite and organic matter in subducted sediments are required to better quantify the impact of this novel mechanism on the global carbon flux.

## Methods
**Investigating the fluid composition in COH-bearing systems**. Several experimental studies investigated separately the effect of the volatiles $H_2O$[35–38] and

$CO_2$[39–42] on subsolidus and melting relations in peridotitic systems at upper-mantle conditions. Only few studies considered the effect of the simultaneous occurrence of $H_2O$ and $CO_2$ or more generally, the influence of COH fluids on the peridotitic systems[3, 43, 44]. The $H_2O/CO_2$ ratio in COH fluids is crucial because it affects the location of carbonation/decarbonation reactions and the position of the solidus. However, the $H_2O/CO_2$ ratio of the fluid in equilibrium with mantle minerals has mainly been estimated through thermodynamic modeling[3], using equations of state of simple $H_2O$–non-polar gas systems[45, 46] (e.g., $H_2O$–$CO_2$–$CH_4$), equations that do not consider the complexity related to dissolution processes[47].

Another fundamental property of $H_2O$-bearing fluids at high-pressure conditions is the capacity to transport dissolved species[18]. The amount of solutes from rock-forming minerals in aqueous fluids increase with increasing pressure[23, 48–50] until the fluid becomes no longer distinguishable from a silicate melt and a supercritical liquid is formed[51, 52]. The solubility of forsterite and enstatite has been measured in pure water up to 1.5 GPa and at $T = 700–1300$ °C[24]. Experimental data on mineral dissolution in mixed $H_2O$–$CO_2$ fluid are available only for quartz[19, 20, 53], albite[54], and diopside[55] and suggests that solubility decreases with increasing content of $CO_2$ in the fluid.

The aim of the paper is to provide for the first time experimental constraints on the composition of high-pressure COH fluids in equilibrium with graphite in terms of both volatile content and dissolved solutes in increasingly complex petrological systems at controlled $fO_2$ conditions. Fluids have been investigated in equilibrium with graphite only in the system COH, with graphite + quartz/coesite in the system $SiO_2$–COH and with graphite + forsterite + enstatite in the system $MgO$–$SiO_2$–COH. Two different experimental setups and analytical techniques were employed to determine the volatile composition and the solubility of minerals in COH fluids at $P = 1.0$ and 3.0 GPa, and $T = 800$ °C. These conditions were selected for sake of simplicity to avoid the presence of carbonates (magnesite) in the system $MgO$–$SiO_2$–COH and the consequent complexities related to carbonate dissolution. In particular, the link between $H_2O/CO_2$ ratio (derived analytically by means of a quadrupole mass spectrometer [QMS][15]) and silicate dissolution (investigated by cryogenic laser-ablation inductively coupled plasma mass spectrometry [LA-ICP-MS][56]) is highlighted here and compared with thermodynamic calculations.

**Bulk compositions and starting materials**. COH fluids were generated starting either from oxalic acid dihydrate (OAD; $H_2C_2O_4 \cdot 2H_2O$) or an equivalent mixture of 1:1 oxalic acid anhydrous (OAA; $H_2C_2O_4$) + water (Supplementary Fig. 1). OAD was employed in the experimental runs aimed at investigating the COH volatile composition by means of the capsule-piercing QMS technique[15] (QMS experiments; Supplementary Fig. 1a–c). The dissociation of oxalic acid at high temperature is given by the reaction:

$$H_2C_2O_4 \cdot 2H_2O \rightleftharpoons 2H_2O + 2CO_2 + H_2, \quad (5)$$

generating a starting fluid characterized by $XH_2O$ ( $= H_2O/CO_2 + H_2O) = 0.5$ and an excess of $H_2$.

Instead of OAD, OAA + Cs-doped (590 µg g⁻¹) water was employed as fluid source in experiments on mineral solubility in COH fluids (LA-ICP-MS experiments; Supplementary Fig. 1d, e), to ensure the presence of an internal standard (Cs) for LA-ICP-MS data quantification[56]. The thermal dissociation of OAA at high-temperature conditions generates a $CO_2$–$H_2$ fluid according to the reaction:

$$H_2C_2O_4 \rightleftharpoons 2CO_2 + H_2. \quad (6)$$

The addition of a proper amount of Cs-doped water in the capsule allowed to obtain a starting fluid composition with $XH_2O = 0.5$ for the LA-ICP-MS experiments too.

Graphite (ceramic-grade powder, checked for purity and crystallinity by X-ray powder diffraction and scanning electron microscopy) was added in all experiments to ensure carbon saturation of the COH fluid. In $SiO_2$–COH and $MgO$–$SiO_2$–COH experiments, natural quartz powder (Supplementary Fig. 1c) and a mixture of synthetic forsterite and enstatite (Supplementary Fig. 1b) were added respectively, with fluid/solid ratios of about 0.2 by weight. Forsterite and enstatite were synthesized from dried nano-crystalline $Mg(OH)_2$ (Sigma-Aldrich, 99.9% purity) and silicon dioxide (Balzers, 99.9% purity), mixed in stoichiometric proportions, pelletized and loaded in a vertical furnace at 1500 °C for 24 h. Synthesis products were ground in ethanol for 1 h, dried and characterized by X-ray powder diffraction analysis (Bruker, AXS D8 Advance, ETH Zurich; Philips X'pert MPD, University of Milan). The resulting composition of the mixture, derived by Rietveld analysis, is forsterite 83.2 wt%, enstatite 16.7 wt%, and cristobalite 0.1 wt%. Run products were characterized using scanning electron microscopy and electron microprobe WDS analyses (Jeol JXA 8200). Silica polymorphs (quartz/coesite) were identified by X-ray diffraction analysis.

In LA-ICP-MS experiments, a layer of diamond powder (grain size 20 µm) serves as a trap for collecting the COH fluids in equilibrium with solids (Supplementary Fig. 1d).

**Experimental conditions and apparatus.** Capsules were welded shut in a frozen steel holder to avoid overheating. Capsules were reweighed to ensure no fluid loss during welding occurred. Sintered MgO rods were employed to embed the capsule, surrounded by a graphite heater, Pyrex glass and NaCl. A rocking piston-cylinder apparatus was used to reach high-pressure and high-temperature conditions. The rocking piston-cylinder is a regular end-loaded piston-cylinder, which allows forward and backward rotation of 180° during the experimental run, thus inverting its position in the gravity field. Chemical inhomogeneity within the capsule is in fact common in fluid saturated experiments. The rocking piston-cylinder overcomes this problem[57] as the rotation of the sample induces Rayleigh–Taylor instabilities, forcing the fluid to migrate and causing chemical re-homogenization. Pressure calibration of the rocking apparatus is based on the quartz to coesite transitions[58] at $P = 3.07$ GPa and $T = 1000$ °C, and $P = 2.93$ at $T = 800$ °C (accuracy $\pm 0.01$ GPa and $\pm 5$ °C). Temperature was measured with a K-type thermocouple located within 0.6 mm from the top of the capsule and is considered accurate to $\pm 5$ °C.

**Buffering strategy and thermodynamic modeling.** Because the volatile composition of graphite-saturated COH fluids is dependent on the redox state of the system, all experiments were performed using a conventional double-capsule design to constrain the chemical potentials (Supplementary Fig. 1).

The outer Au capsule (OD = 5 mm in QMS experiments; OD = 3 mm in LA-ICP-MS experiments) contains a buffering mixture of either Ni + NiO + $H_2O$ (NNO) or fayalite + magnetite + quartz + $H_2O$ (FMQ; ferrosilite + magnetite + coesite at 3 GPa, 800 °C). As long as all phases are present in the buffering mixtures, they fix the fugacity of $H_2$ ($fH_2^{NNO}$, $fH_2^{FMQ}$) through the reactions:

$$Ni + H_2O \rightleftharpoons NiO + H_2, \tag{7}$$

$$3Fe_2SiO_4 + 2H_2O \rightleftharpoons 2Fe_3O_4 + 3SiO_2 + 2H_2. \tag{8}$$

The inner $Au_{60}Pd_{40}$ capsule (OD = 2.3 mm), which contains the COH fluids in equilibrium with graphite $\pm$ quartz/coesite $\pm$ forsterite + enstatite, is permeable to hydrogen. Therefore, the fugacity (and, thus, the chemical potential) of $H_2$ is expected to be homogeneous and identical in the inner and in the outer capsules. Indirectly, also all the other species composing the COH fluid will be externally buffered, including oxygen. However, since the inner capsule contains a mixed COH fluid instead of pure water, the oxygen fugacity in the inner capsule will be lower compared to that fixed in the outer capsule. The fugacities of oxygen and hydrogen fixed in the outer capsule by both NNO and FMQ were calculated employing Perple_X[59] (http://www.perplex.ethz.ch) using the thermodynamic data set of Holland and Powell[60] revised by the authors in 2004 and the Perple_X EoS n. 16 (H–O HSMRK/MRK hybrid EoS) (Supplementary Table 2). Subsequently, we calculated the speciation of the graphite-buffered COH fluid in the inner capsule for $fH_2$ fixed by both NNO and FMQ through thermodynamic modeling using (i) the software package Perple_X[59] (based on Gibbs free energy minimization) and the EoS of Connolly and Cesare[45] (Perple_X EoS n. 11) (CC93 in Supplementary Table 2); (ii) the Excel spreadsheet GFluid[31] (based on gaseous equilibrium constants $K_p$) with the EoS of Zhang and Duan[46] and the static $H_2$ fugacity coefficient provided in the spreadsheet (ZD09 in Supplementary Table 2); (iii) the Excel spreadsheet GFluid[31] (based on gaseous equilibrium constants $K_p$) with the EoS of Zhang and Duan[46], with a modified $H_2$ fugacity coefficient ($\gamma H_2$) changing dynamically as a function of $X(O)$, fitted from the EoS of Connolly and Cesare (1993) ($\gamma H_2 = a \bullet X(O)^3 + b \bullet X(O)^2 + c \bullet X(O) + d$, where: at 1 GPa and 800 °C, $a = -43.919$, $b = 114.55$, $c = -105.75$ and $d = 41.215$; at 3 GPa and 800 °C, $a = -11,208$, $b = 26723$, $c = -21,949$ and $d = 6979.2$) (ZD09mod in Supplementary Table 2). By assuming that $fH_2$ of the COH fluid in the inner capsule is equal to $fH_2^{NNO}$ and $fH_2^{FMQ}$, we were able to calculate the molar fractions of volatiles ($H_2O$, $CO_2$, CO, $CH_4$, $H_2$, $O_2$), the $X(O)$, the $fO_2$ and the $\Delta FMQ$ (= log $fO_2$ = -log $fO_2^{FMQ}$) of the COH fluid in the inner capsule at the investigated $P$ and $T$ conditions (Supplementary Table 2 for modeling at 1–3 GPa and 800 °C). Depending on the thermodynamic models used, predicted fluids are characterized by the following $\Delta FMQ$ values:

−0.67 (CC93), −0.64 (ZD09) and −0.58 (ZD09mod) at 1 GPa, 800 °C, $fH_2^{NNO}$;
−0.73 (CC93), −0.70 (ZD09) and −0.61 (ZD09mod) at 1 GPa, 800 °C, $fH_2^{FMQ}$;
−0.42 (CC93), −1.17 (ZD09) and −0.47 (ZD09mod) at 3 GPa, 800 °C, $fH_2^{FMQ}$.

Predicted fluids are mainly composed of $H_2O$ and $CO_2$, with $XCO_2$ [ = $CO_2/(CO_2 + H_2O)$] generally higher compared to the starting equimolar $H_2O$–$CO_2$ composition (Supplementary Table 2). In fact, the equilibration of the COH fluid in the inner capsule is accomplished by these coupled reactions:

$$H_2O \rightleftharpoons H_2 + \tfrac{1}{2}O_2, \tag{9}$$

$$C + O_2 \rightleftharpoons CO_2, \tag{10}$$

which can be condensed to the following graphite-consuming reactions:

$$C + 2H_2O \rightleftharpoons CO_2 + 2H_2, \tag{11}$$

$$C + 2H_2 + 2O_2 \rightleftharpoons CO_2 + 2H_2O. \tag{12}$$

In conclusion, the equilibration of the COH fluid at run conditions implies that $CO_2$ is produced in the inner capsule by oxidation of graphite. As a consequence, the $XCO_2$ of the COH fluid in the inner capsule increases until equilibrium in $fH_2$ is reached between the inner and outer capsules (Supplementary Fig. 4).

We experimentally verified the fluid composition predicted by thermodynamic modeling, by retrieving analytically the $XCO_2$ ratio by means of QMS technique[15], summarized in the next section. At both $P = 1$ GPa, $T = 800$ °C and $P = 3$ GPa, $T = 800$ °C measured and calculated ratios are concordant within errors with ZD09mod for both NNO- and FMQ-buffered experiments performed in the COH system (Supplementary Tables 1 and 2; Supplementary Fig. 2). Therefore, we conclude that this model can reproduces experimental data in the pure COH system. In the $SiO_2$–COH and MgO–$SiO_2$–COH systems, literature data and the new experimental data discussed in this paper demonstrate that dissolved Mg-bearing and Si-bearing species resulting from dissolution reactions taking place between solid phases and COH fluids occur in addition (Fig. 2; Supplementary Table 3). By retrieving the $XCO_2$ in these systems analytically using the capsule-piercing QMS technique[15], we have demonstrated that the thermodynamic calculations outlined above, which do not take into account these dissolved species, cannot be applied to predict the fluid composition in complex systems, in particular those bearing silicates.

The aqueous speciation and solubility calculations in the investigated silicate-bearing systems were carried out using mass balance, charge balance, and mass action expressions in the code EQ3[26]. The equilibrium constants involving aqueous ions, water, and minerals were calculated using the DEW model[27]. Aqueous ionic activity coefficients were calculated using the extended Debye–Hückel equation including the conversion from the mole fraction scale to the molality scale. The activity coefficients of neutral aqueous $CO_2$ and $CH_4$ were approximated in the MgO–$SiO_2$–COH system by a conversion of the standard states in the model for COH fluids of Zhang and Duan[31] to the hypothetical 1.0 molal standard state. The activity of $H_2O$ was approximated in the MgO–$SiO_2$–COH system by its mole fraction. Additional new equilibria for the species $Mg(OH)_{2(aq)}$ and $Mg[OSi(OH_3)]$ $[CH_3CH_2COO]^0$ (Supplementary Table 4) were fitted to the experimental solubilities in the systems MgO–$SiO_2$–$H_2O$ and MgO–$SiO_2$–COH, respectively at 800 °C and 1.0 GPa. The calculated molalities of the most abundant Mg- and Si-species are given in Supplementary Table 4. It can be seen from the model results that both the $Mg(OH)_{2(aq)}$ and $Mg[OSi(OH_3)][CH_3CH_2COO]^0$ species are predicted to contribute significantly to the total solubilities of both Mg and Si in the MgO–$SiO_2$–COH system. However, because of the low activity of water in the system the amounts of $Mg(OH)_{2(aq)}$, $Si(OH)_{4(aq)}$, and $(OH)_3SiOSi(OH)_{3(aq)}$ are much lower than in the carbon-free system. Under more oxidizing conditions, the $Mg[OSi(OH_3)][CH_3CH_2COO]^0$ species would become unimportant, and the solubilities of Mg and Si in the MgO–$SiO_2$–COH system could become extremely low.

**Analysis of volatiles.** Here, we provide a brief summary of the technique used to measure the volatile composition of the COH fluids in the inner capsule. The full description and the validation of the technique is provided in Tiraboschi et al.[15] Quenched COH fluids are extracted from the inner capsule and conveyed to a quadrupole mass spectrometer (QMS). The capsule-piercing device consists of an extraction vessel (reactor) that is heated by an electric furnace to $T = 80$ °C to transform liquid water into water vapor. The reactor, made of Teflon, is composed of a basal part, where the capsule is placed, and a top part, where a steel pointer is mounted. The piercing is executed by screwing the basal part on to the top part until the pointer penetrates the capsule. The capsule, partially embedded in epoxy, is mounted on a steel support, designed to oppose the rotation exerted by the steel pointer during the piercing operation. The furnace design includes a pilot hole in the base part that permits screwing the reactor with a hex key while placed in the furnace. Openings on the top of the reactor allow the carrier gas (ultrapure Ar) to flow inside it and to generate vent or vacuum conditions. The presence of O-rings ensures a tight seal. The reactor is connected to a QMS by a heated line (80 °C) to avoid condensation of water on the metal tubes. The pressure conditions in the line and in the reactor are monitored through high-resolution sensor gauges ($\pm 1$ mbar precision). The temperatures of the line, reactor and furnace are monitored with K-type thermocouples. Line and reactor pressures and temperatures are recorded by a Eurotherm Nanodac data recorder with PID control. The internal volume of the reactor was calibrated against the distance $h$ between the basal and the top part. $P$–$T$–$V$ conditions in the reactor at the time of piercing are used to retrieve the total moles, $n$, of gases released from the capsule, following the ideal gas law $n = RT/PV$.

For the calibration of the QMS, standard gas mixtures of known compositions were utilized: (i) 80 vol% Ar, 10 vol% $CO_2$, 10 vol% $O_2$; (ii) 80 vol% Ar, 10 vol% $CH_4$, 10 vol% CO; and (iii) 90 vol% Ar, 10 vol% $H_2$. Water calibration is performed by loading a known amount of water (typically 1 µl) with a micro-syringe inserted through a silicon septum present at the top into the reactor. The calibration allows

performing quantitative analyses of $H_2O$, $CO_2$, $CH_4$, $CO$, $H_2$, and $O_2$. Uncertainties for major species were typically ~1% for $H_2O$ and $CO_2$, and ~10% for $CO$. After piercing has occurred, gases are conveyed to the QMS by opening a valve. For every $m/z$ channel, the QMS counts are measured every 5 s for 310 steps, for a total of 1550 s of measurement time. The moles of gases were obtained by comparing the areas of the $m/z$ peaks with those of the standards, using a least-squares regression method. Monte Carlo simulations provided the propagation of uncertainties for each species, which corresponds to measurement uncertainty of the sample and can be represented as a probability distribution plot in ternary COH diagrams.

**Analysis of solutes**. The solute content in the fluid was measured employing the cryogenic LA-ICP-MS technique also known as the "freezing technique"[56], which is applied for the first time on double capsules bearing COH fluids. The recovered experimental capsule is mounted on a freezing stage, which consists of a stack of two Peltier elements, surrounded by plastic to thermally insulate the elements from the atmosphere[61]. The sample holder is placed on a copper block, in direct contact with the Peltier elements and cooled to $T \sim -35\,°C$. The conventional freezing technique has been updated using a cutter blade mounted on a steel support. This device allows cutting longitudinally double capsules by fastening a screw that pushes the cutter blade via a steel block through the capsule. During this operation, the capsule is enclosed in a copper holder. Once the capsule is cut open, the cutting device is removed from the freezing stage together with the upper part of the capsule holder including the top part of the capsule. The upper half of the capsule is investigated using a binocular microscope, while the lower part remains frozen on the stage during the entire laser-ablation analytical session that follows immediately.

Analyses were performed using a 193 nm ArF GeoLas Pro excimer laser system coupled to an ELAN DRCD-e quadrupole mass spectrometer at University of Bern. We analyzed the diamond trap for $^{24}Mg$, $^{25}Mg$, $^{26}Mg$, $^{29}Si$, $^{62}Ni$, $^{133}Cs$, $^{195}Pt$, and $^{197}Au$, using a 60 μm beam diameter, ~13 J/cm laser fluency, and 5 Hz repetition rate. Data were acquired in blocks of up to ~ 10 individual sample analyses bracketed by three analyses of the standard NIST SRM610, placed in the ablation chamber with the sample. Background was taken for ~50 s and the sample signal, on the diamond trap or on the solid residue, was collected for ~20 s. LA-ICP-MS data reduction employed the SILLS software[62] and in-house spreadsheets to calculate solute concentrations, employing rigorous limit of detection filtering[63] for each element and each measurement individually.

The cryogenic technique has been originally developed for analyzing the solute content of aqueous fluids. Cesium, introduced in the starting materials, is employed as an internal standard for data quantification, because it is a highly incompatible element that fractionates completely into the aqueous fluid phase at the given residual mineralogy. In our experiments, we introduced a known amount of water solution doped with 590 μg g$^{-1}$ Cs [as $Cs(OH)_2$]. As the initial $Cs/H_2O$ ratio is fixed, once the Cs concentration in the fluid coexisting with minerals at run $P$ and $T$ is known, solute concentrations of the fluid can be calculated[56]. However, compared to experiments bearing aqueous fluids, our double-capsule, COH-bearing experiments are more complex, because the initial $Cs/H_2O$ is not fixed, as the water content in the inner capsule is variable, depending on $P$, $T$ and $fH_2$–$fO_2$ conditions (Eqs. 11 and 12). In fact, in double-capsule arrangements $H_2$ is a mobile component that can be added or removed from the system through diffusion in and out of the inner capsule. This implies that the initial Cs concentration cannot be used as an internal standard. This value needs to be corrected taking into account the change in total water mass present in the inner capsule relative to initial amount of loaded water, as a consequence of fluid $XCO_2$ re-equilibration at the $fH_2$–$fO_2$ conditions imposed by the buffers. If $H_2O$ is consumed during fluid re-equilibration at run conditions, Cs concentration in the residual water increases; if $H_2O$ is produced, Cs concentration decreases (Supplementary Fig. 3). We estimated the corrected Cs concentration at run $P$ and $T$ using a model, which assumes that fluid equilibration at the hydrogen fugacity conditions imposed by the buffers NNO and FMQ is governed only by $H_2$ mobility and no hydration or carbonation reactions occur in the capsule charge. As long as these two assumptions are valid, it is possible to estimate the amount of Cs in the inner capsule in the following way considering a classic dilution equation:

$$C_iCs \times V_iH_2O = C_fCs \times V_fH_2O, \tag{13}$$

where $C_iCs$ is the initial concentration of Cs in the aqueous solution loaded into the capsule (590 μg g$^{-1}$) and $C_fCs$ is the final concentration of Cs after fluid equilibration at $fH_2$ conditions. $V_iH_2O$ and $V_fH_2O$ are the initial and final volumes of water.

The volume of water is proportional to the moles according to:

$$VH_2O = nH_2O \times V_{mol}H_2O, \tag{14}$$

($VH_2O$, volume of water; $nH_2O$, number of molesof water; $V_{mol}H_2O$, molar volume of water).

Considering that at fixed pressure and temperature conditions the molar volume of water is constant, we obtain the following dilution equation:

$$C_iCs \times n_iH_2O = C_fCs \times n_fH_2O. \tag{15}$$

The final Cs concentration in the aqueous fluid fraction will be given by:

$$C_fCs = \frac{C_iCs \times n_iH_2O}{n_fH_2O}. \tag{16}$$

As $n_iH_2O$ is known (the initial amount of water charged into the capsule), this equation can be solved as long as $n_fH_2O$ is constrained, through equation:

$$n_fH_2O = n_iH_2O \times \frac{XH_2O_f}{XH_2O_i}, \tag{17}$$

where $XH_2O_f$ is the final $H_2O/(CO_2 + H_2O)$ ratio measured by QMS analysis, and $XH_2O_i$ is the initial $H_2O/(CO_2 + H_2O)$ ratio retrieved on the basis of the amounts of water and OAA charged into the capsule. Solute concentrations in the aqueous fraction of the COH fluid at run $P$ and $T$ can thus be calculated (Fig. 2; Supplementary Table 3).

**Data availability**. The authors declare that the data supporting the findings of this study are available within the article.

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

## Acknowledgements

Authors are indebted to A. Risplendente for the assistance at scanning electron microscope and electron microprobe. M. Merlini is acknowledged for the characterization of the experimental materials by X-ray diffraction. Funding provided by the Italian Ministry of Education, University and Research (MIUR) program PRIN2012R33ECR. S.T., C.T., D.S. and S.P. acknowledge support from the Deep Carbon Observatory (DCO).

## Author contributions

The capsule-piercing apparatus was conceived by S.T. and built by S.R. QMS analyses were performed by S.T., S.R., C.T., and F.M. LA-ICP-MS analyses were carried out by T.P., P.U., C.T., and S.T. Thermodynamic modeling was performed by S.T. and D.S. All authors participated in extensive discussions and the preparation of the manuscript.

## Additional information

**Competing interests:** The authors declare no competing financial interests.

