## [Peer Review File · Nature Communications]

Reviewers' Comments:

Reviewer #1 (Remarks to the Author)

Review of manuscript

Silicate dissolution boosts the CO₂ concentrations in subduction fluids

Authors:

S. Tumiatì, C. Tiraboschi, T. Pettke, S. Recchia, P. Ulmer and S. Poli

Submitted to Nature Communications

General comment

This well-written and scientifically sound work addresses a very important topic, i.e. the transport properties of fluid solutions, in this case the solubility of CO₂ in silica and magnesium-rich fluids. The experimental approach, the technical aspects of which are also quite interesting and of high quality, focuses on a geodynamic site where mixed Si-Mg-rich fluids might be expected: the slab-mantle interface in subduction zones and thus yields enormous information on the CO₂ cycling in this important tectonic setting. Although, I wonder whether the experimental conditions (800-900°C at 1 GPa) reflect natural conditions in the envisaged site, the results of this study are most important for a large geoscientific community, the arguments of the authors' are well supported by the results and are worthwhile publishing in a high impact journal.

Nevertheless, I have some minor concerns regarding immediate publication in Nature Communications. An important point that I am missing is a discussion about the global impact of the findings, such as a comparison with previous estimates of CO₂ cycling in subduction zones (e.g., Gormann et al., 2006; Connolly 2005). Such a discussion would make the manuscript suitable for a broader geoscientific community. How important are the results for the global CO₂ budget especially in comparison to the relation of natural/anthropogenic CO₂ emissions?

We have seen in other publications, such as the work of Craig Mannings group, that complexing in mixed volatiles, e.g., in the albite-supercritical fluid/hydrous melt system, immensely increases the solubility of different species. Hence, this effect is already known and thus the provocative/surprising aspect of this manuscript is slightly abated. Nevertheless, this circumstance does not lessen the importance of this study, but the manuscript would benefit from putting the results into a global context.

However, it is obvious that experimentally derived (e.g., thermodynamic) datasets usually are very important, tedious to derive and often the results of such studies solely increase the number of entries in a thermodynamic database and will never get cited again. Therefore I generally support publication of such experimental works in high-impact journals, such as Nature Communications. It has, however, to be admitted that purely experimental studies seem to be somewhat bland and dreary to read for people who do not share the topic. Thus I suggest adding a small paragraph regarding the global impact of these very interesting and important findings. After these minor revisions the manuscript is suitable for publication in Nature Communications.

I hope my comments are helpful to improve the impact of that well-written and interesting manuscript.

References cited in this review

Gorman, Patrick J., D. M. Kerrick, and J. A. D. Connolly. "Modeling open system metamorphic decarbonation of subducting slabs." *Geochemistry, Geophysics, Geosystems* 7.4 (2006).

Connolly, J. A. D. "Computation of phase equilibria by linear programming: a tool for geodynamic modeling and its application to subduction zone decarbonation." *Earth and Planetary Science Letters* 236.1 (2005): 524-541.

Reviewer #2 (Remarks to the Author)

This manuscript describes experimental results suggesting that at high pressure much more CO₂ is dissolved by fluids from graphite bearing rocks than is predicted by models. The higher carbon content correlates with higher than expected solute contents (Si and Mg). If correct the result is significant because it would suggest that much more carbon can be lost from subducting rocks than we previously thought, contributing to the emerging debate about the deep carbon cycle.

While the experiments seem sound, the explanation for what is happening is inadequate. The correlation of elevated CO₂ with elevated solutes would be consistent with the reactions written on page 5 if all we knew about was the MS-COH experiments on forsterite+enstatite. However, the same behavior is seen in quartz-COH experiments, which poses a problem for this explanation.

The authors note that the distribution of species in the buffer will differ from that in the charge due to dissolved solutes in the latter. It is possible that this can be part of the explanation, but the authors do not attempt to quantify this. That's puzzling because the chemical potentials can be (and were) calculated for the pure system, so one could simply apply these to the inner capsule to assess the magnitude of the difference between required and observed compositional effects to see if all this could possibly be explained by dissolved solutes. I doubt it, the total dissolved solutes are not actually that large and the effect of hydration of solutes on H₂O activity is probably not large enough. I'd be happy to be proved wrong on this but the point is it is part of the necessary path to providing a more concrete and less speculative explanation.

It is also noteworthy that new thermodynamic data exist for the speculated carbonate and Mg-carbonate species (Sverjensky et al 2014). In the same spirit as the above suggestion, one could calculate whether the thermodynamic data indeed predict the observed concentrations. This is akin to what Galvez et al seem to have done, though obviously for a more complex system.

In sum, the experiments are intriguing and potentially important, but I cannot recommend publication of this manuscript without a clearer and more quantitative thermodynamic explanation for the observed discrepancy.

Reviewer #3 (Remarks to the Author)

The manuscript by Tumiatti et al. titled "Silicate dissolution boosts the CO₂ concentrations in subduction fluids" reports measurements of quenched fluid from high pressure experiments using a newly developed technique. Here the authors measure compositions of H₂O-CO₂ fluids generated at graphite saturated conditions and also in the presence of silicate minerals (forsterite, enstatite, or quartz); in particular, the authors measure the concentration of 'dissolved' solute such as SiO₂ and MgO in addition to the molecular gas species. The main finding is that the CO₂ dissolution seems to increase with the presence of dissolved solutes.

The authors' effort to measure quenched fluid using capsule piercing technique and fluid measurements using quadrupole mass spectrometry is interesting. However, I find the work and the suggested implications not so convincing for the following reasons -

- 1) The conditions of experiments, which is 1 GPa and 700-1000 °C is not directly applicable to carbon transfer in subduction zone, as the depth is much shallower than typical sub-arc depths. The authors may claim that their study suggests that significant CO₂ transfer from graphite-saturated lithologies would occur even at shallower depths, but no study has found significant CO₂ flux at forearc conditions so far. So the question is how reliable are these measurements or how relevant are these.
- 2) It is surprising that the authors focused on forsterite and enstatite as the two main residual minerals in the presence of graphite. If graphite is present then it is most likely in the sediments or may be in the basaltic crust and not in the overlying mantle wedge. The main CO₂ transfer from slab to mantle wedge definitely initiates in the crust or the sediment and not in the mélange zone.

The reason this is critical is that the solute complexes in the fluid depend critically on the residual minerals; it is certain that the solute-fluid complexes would be different if various aluminous phases such as garnet, feldspar, kyanite, omphacite etc. are considered and these are surely more important residual minerals to consider other than quartz in subducting crustal assemblages. The question is would the CO₂ dissolution be enhanced as the authors suggest if aluminous complexes are formed in the fluid (e.g., Mookherjee et al., 2015 - GCA). If CO₂ dissolution in silicate melt is any hint, the answer would be 'no'. More aluminous melt tends to suppress CO₂ solubility.

3) It is also unclear why the authors focused on CO₂ transfer in fluid when it is saturated in graphite rather than carbonate minerals. Hardly any justification is given for that. In modern subduction zones, the main mode of carbon subduction is through subduction of mineral carbonates. So shouldn't one be more interested in knowing CO₂ content of silicate minerals and carbonate-saturated fluids and characterizing the solute species in such fluids?

4) The technique - I am also not sure that the analysis of MgO and SiO₂ solute in fluid using cryogenic laser ablation ICP-MS is necessarily giving an accurate measurement. The problem is, as the authors note, that the fluid is exsolved into multiple phases during quench and the authors assume that all the solute partitions into the aqueous part and not to the CO₂-rich part. But how do we know that is the case?

Finally, I found the manuscript very difficult to read. It is very dense and has a structure not very useful to convey an important message in global carbon cycle. The authors made the manuscript too technical. Of course the decision on the manuscript should be an editorial decision, but I did not find the work to be of broad interest but rather appropriate for scrutiny in a technical journal.

Comments by the three reviewers are most welcome as they provide thoughtful suggestions helpful to arrange this revised version. Below, we enumerated responses to referees' comments. Although the evaluation by the Reviewer #3 was negative, we appreciated that he acknowledges the main point of the manuscript: "*The authors' effort to measure quenched fluid using capsule piercing technique and fluid measurements using quadrupole mass spectrometry is interesting*". His severe criticism forced us to run **new experiments**, to **reorganize the manuscript** and to **include new sections, figures and tables in the main text and in Supplementary Information**.

We mostly agree with issues raised by the reviewers and, after all changes we propose, we hope we will be able to convince the reader about the robustness of the data presented here and of the unexpected role of silica in boosting the CO₂ content in graphite-saturated fluids, with implications for the global carbon flux.

Reviewer 1

He stressed out the importance and the quality of our data, in particular concerning the "*transport properties of fluid solution*" flushing a subduction mélange. The minor concerns of Reviewer 1 have been addressed as follows:

1) An important point that I am missing is a discussion about the global impact of the findings, such as a comparison with previous estimates of CO₂ cycling in subduction zones (e.g., Gormann et al., 2006; Connolly 2005)... I suggest adding a small paragraph regarding the global impact of these very interesting and important findings.

Our experimental data demonstrate that independently from the occurrence of carbonates, the dissolution of silicates can boost the dissolution of graphite in the subduction mélange in the form of volatile CO₂ dissolved in COH fluids by up to + 30 % compared to systems not bearing silicates. Most of the estimates of the CO₂ cycling in subduction zones take into account the contribution of carbonates, but only few papers dealt with the role of organic carbon and/or the graphite content in subducted rocks. Kelemen & Manning (2015) adjusted the contribution of subducted carbon-bearing sediments retaining the lower-bound flux of 13 Mt C/y (due to sedimentary carbonates) but adjusting the upper bound from 17 Mt C/y to 23 Mt C/y, including in their model also the sedimentary organic carbon. Therefore, available thermodynamic models predict that the contribution of sedimentary organic carbon should account for at least 6 (=23-17) Mt C/y. We suggest that this model underestimates the real contribution of organic carbon to the global carbon flux by 10–30%, because experimental data show that fluids interacting with silicates are more effective in prompting the oxidation of graphite to CO₂.

We added a small paragraph in the section *Dissolution of graphite and silicates in subduction mélanges*:

"Moreover, because this CO₂ boosting effect cannot be predicted by available thermodynamic models, and these have been used to estimate the amount of CO₂ recycled from subducted, carbon-bearing sediments⁸, the estimated carbon transfer linked to the oxidation of sedimentary organic carbon and graphite (~ 6 Mt C/y) probably needs to be adjusted up by 10–30% (0.6–1.8 Mt C/y). Other experiments at different *P*, *T*, *X* conditions are needed in order to better quantify the impact of this novel mechanism on the global carbon flux"

2) *We have seen in other publications, such as the work of Craig Mannings group, that complexing in mixed volatiles, e.g., in the albite-supercritical fluid/hydrous melt system, immensely increases the solubility of different species. Hence, this effect is already known and thus the provocative/surprising aspect of this manuscript is slightly abated.*

Experimental data concerning the solubility of fluids in pure water are relatively abundant, but experimental solubility data in mixed H₂O-CO₂ fluids are very few. More important, no one suggested before that solutes are able to modify the carbon content of COH fluids. Instead, CO₂ was considered to be an "inert" diluent. **We believe that the effect of dissolved silica in boosting the CO₂ content of COH fluids is really stunning and it can not be predicted by available thermodynamic models, so a completely novel effect is described in our manuscript.** In order to make our data even stronger, we decided to replicate all the experiments, both those used for the determination of the volatile content by means of quadrupole mass spectrometry and those used for the determination of solutes by means of laser-ablation ICP-MS. Moreover, **we performed new experiments buffered by fayalite-magnetite-quartz (FMQ)** in order to check if the claimed increase in XCO₂ will occur in the same systems buffered at different $f_{\text{H}_2}/f_{\text{O}_2}$. **All the results presented in the former version of the manuscript have been confirmed.**

Reviewer 2

We appreciated that Reviewer 2 finds our data "*intriguing and potentially important*". He poses some important questions about possible explanations of the experimental results we found. We hope that the new experiments and the new thermodynamic models we added in the manuscript can convince the Reviewer 2 about the robustness of our data.

1) *While the experiments seem sound, the explanation for what is happening is inadequate. The correlation of elevated CO₂ with elevated solutes would be consistent with the reactions written on page 5 if all we knew about was the MS-COH experiments on forsterite+enstatite. However, the same behavior is seen in quartz-COH experiments, which poses a problem for this explanation.*

We agree with the reviewer. The former reactions involving carbonate and bicarbonate ions are inadequate to explain the observed boosting effect, because theoretical model results indicate that at 1.0 GPa and 800° C no significant amounts of HCO₃⁻ or CO₃²⁻ are present in the fluids. Therefore, **we performed thermodynamic models considering the EoS of Zhang & Duan (2009) in addition to the EoS of Connolly & Cesare (1993), for COH fluids in equilibrium with graphite alone. For silicate bearing systems, we used the aqueous speciation-solubility code EQ3 adapted to include equilibrium constants calculated with the Deep Earth Water model.**

2) *The authors note that the distribution of species in the buffer will differ from that in the charge due to dissolved solutes in the latter. It is possible that this can be part of the explanation, but the authors do not attempt to quantify this. That's puzzling because the chemical potentials can be (and were) calculated for the pure system, so one could simply apply these to the inner capsule to assess the magnitude of the difference between required and observed compositional effects to see if all this could possibly be explained by dissolved solutes. I doubt it, the total dissolved solutes are not actually that large and the effect of hydration of solutes on H₂O activity is probably not large enough. I'd be happy to be proved wrong on this but the point is it is part of the necessary path to providing a more concrete and less speculative explanation.*

It is also noteworthy that new thermodynamic data exist for the speculated carbonate and Mg-carbonate species (Sverjensky et al 2014). In the same spirit as the above suggestion, one could calculate whether the thermodynamic data indeed predict the observed concentrations.

We found that the measured volatile composition of fluids in equilibrium with graphite alone **can**

be reproduced almost perfectly using the EoS of Zhang & Duan (2009), if the H₂ fugacity coefficient of Connolly & Cesare (1993) is considered. We were able to reproduce also the measured dissolved silica using the speciation-solubility model, but this model fails in reproducing the observed increase in CO₂ in fluids in equilibrium with silicates. **The increase in CO₂ can be modeled using the modified EoS of Zhang & Duan (2009) by either increasing the f_{O_2} or decreasing the f_{H_2O} by 0.03-0.06 log units.** Therefore, we argue that dissolved silica is effective in decreasing water activity and/or increasing oxygen fugacity, and consequently, it boosts the CO₂ content of high pressure fluids, provided that graphite is present in excess, independently from the occurrence of carbonates. **Dimitri Sverjensky** helped in doing thermodynamic modeling and he is now an author of the manuscript.

Reviewer 3

Although the evaluation by the Reviewer #3 was rather negative, we appreciated his constructive criticism.

1) The conditions of experiments, which is 1 GPa and 700-1000 {degree sign}C is not directly applicable to carbon transfer in subduction zone, as the depth is much shallower than typical sub-arc depths. The authors may claim that their study suggests that significant CO₂ transfer from graphite-saturated lithologies would occur even at shallower depths, but no study has found significant CO₂ flux at forearc conditions so far. So the question is how reliable are these measurements or how relevant are these

Massive diffuse outgassing at forearc conditions has been reported in recent literature (e.g., Kelemen & Manning, 2015). **P , T condition of 1 GPa and 800° C has been chosen on order to avoid the presence of carbonates** (magnesite) in the MS-COH system and to add constraints to previously published studies investigating at similar P , T conditions the solubility of carbon and silicates in high pressure aqueous fluids. Actually, once graphite and carbonates occur together, it becomes almost impossible to establish the role exerted by each mineral phase on CO₂ concentration. More complex experiments will be performed in the next future.

In the main text:

"A carbonate-free compositional range has been explored at $P = 1$ GPa and $T = 800$ °C in order to focus on the role of graphite and silicates in the investigated processes"

Supplementary information:

"These conditions were selected for sake of simplicity to avoid the presence carbonates (magnesite) in the system MS-COH and the consequent complexities related to carbonate dissolution"

2) It is surprising that the authors focused on forsterite and enstatite as the two main residual minerals in the presence of graphite. If graphite is present then it is most likely in the sediments or may be in the basaltic crust and not in the overlying mantle wedge. The main CO₂ transfer from slab to mantle wedge definitely initiates in the crust or the sediment and not in the mélange zone

We focused on the upper interface of the subduction zone, where the sedimentary cover of the subducting slab is in contact with the overlying mantle wedge. Therefore, in this setting sedimentary rocks and mantle rocks are the most common lithologies found in the mélange. In order to take into account the subducted crust, more complex systems with many variables have to be investigated, including Na, Al, Fe, Ca... We preferred to investigate a relatively simple but well constrained system first, extending the compositional range in the next future.

Fluids coming from the dehydration of the slab are thought to be rich in H₂O and poor in CO₂, because carbonated are very stable at HP conditions. Decarbonation is not thought to be a common process in subduction settings, and CO₂ removal has been recently explained through dissolution of carbonates occurring in altered oceanic lithosphere and its sedimentary cover, diapirism of slab rocks and/or melts, but these processes are far from being completely understood.

3) The reason this is critical is that the solute complexes in the fluid depend critically on the residual minerals; it is certain that the solute-fluid complexes would be different if various aluminous phases such as garnet, feldspar, kyanite, omphacite etc. are considered and these are surely more important residual minerals to consider other than quartz in subducting crustal assemblages. The question is would the CO₂ dissolution be enhanced as the authors suggest if aluminous complexes are formed in the fluid (e.g., Mookherjee et al., 2015 - GCA). If CO₂ dissolution in silicate melt is any hint, the answer would be 'no'.

Simply we cannot answer this questions now, because experimental data are not available. We observed that dissolved silica is much more effective than dissolved magnesium in boosting the XCO₂ of the fluid in equilibrium with graphite, probably because silica affects water activity in an unpredicted manner. So, if aluminum behaves like magnesium, it will not influence substantially the CO₂ content of COH fluids. This has to be checked experimentally.

4) It is also unclear why the authors focused on CO₂ transfer in fluid when it is saturated in graphite rather than carbonate minerals. Hardly any justification is given for that. In modern subduction zones, the main mode of carbon subduction is through subduction of mineral carbonates. So shouldn't one be more interested in knowing CO₂ content of silicate minerals and carbonate-saturated fluids and characterizing the solute species in such fluids?

Experiments performed with graphite have some advantages, for example their redox state can be modeled using EoS of graphite-saturated fluids (Connolly & Cesare, 1993; Zhang & Duan, 2009). The great value of our experiments rely on the use of double capsules to synthesize carbon- and redox-buffered COH fluids, which were measured for their volatile and solute contents. It is certainly possible to add carbonates to the investigated systems, to check if dissolution of carbonates can affect the XCO₂ of the graphite-saturated fluid, but these will be more complex experiments that most likely would not be understood without the constraints that we have got in the investigated more simple systems, where the number of variables is lower.

5) The technique - I am also not sure that the analysis of MgO and SiO₂ solute in fluid using cryogenic laser ablation ICP-MS is necessarily giving an accurate measurement. The problem is, as the authors note, that the fluid is exsolved into multiple phases during quench and the authors assume that all the solute partitions into the aqueous part and not to the CO₂-rich part. But how do we know that is the case?

In the MS-H and S-COH systems, our experimental LA-ICP-MS data are comparable to data retrieved using the weight-and-loss technique and to values obtained by thermodynamic speciation-solubility modeling. We can exclude the occurrence of precipitates in the runs, which were checked at the electron scanning microscope.

6) Finally, I found the manuscript very difficult to read. It is very dense and has a structure not very useful to convey an important message in global carbon cycle. The authors made the manuscript too technical.

We made efforts in order to re-arrange the text and make it more readable. Technical details are

available as Supplementary Information.

-

Reviewers' Comments:

Reviewer #1 (Remarks to the Author)

One of my major criticism about publication of the manuscript "Silicate dissolution boosts the CO₂ concentrations in subduction fluids" by Tumiati et al. was that the focus of the work, or at least the discussion/conclusion was too much on the geochemistry/technical results without setting them into a "global" context. Here I fully agree with reviewer #3, who also stated that point. The authors, however, decided to address this point by adding a small paragraph about the impacts on the global carbon cycle, without changing the overall character of the manuscript. Therefore, the text reads now a little better and if you want you can imagine what the major impact of that work on global element cycles is, but I am still not convinced that a broad audience would get that point. For example, how significant in terms of the "natural" vs. anthropogenic CO₂ budget and the carbon cycle is a 30% increase in carbon transfer from the subducted plate? For geochemists/petrologists/geodynamicists this is of course of fundamental relevance, but I think that Nature Communications is aiming for a broader audience.

Reviewer #2 (Remarks to the Author)

This manuscript describes an intriguing set of observations from experiments on quartz and forsterite+enstatite in the presence of graphite and COH fluid at high temperature and pressure. There are two novel insights. The first is that there is higher solubility of Mg silicates in the COH fluid than in pure H₂O, which is unexpected. The second is that the CO₂ content of the fluid is higher than it should be if the only consideration is graphite saturation at a known fH₂ and fO₂.

The revised manuscript has been significantly modified to address my earlier complaint that it is necessary to provide/propose a mechanism for these observations. However, the explanations for the two surprises are in each case speculative as the manuscript stands now. Both could be made quantitative, and therefore better support the hypotheses. In my view, this step is necessary to make it publishable in Nature Communications over a regular geochemistry journal.

1. Enhanced solubility of Mg silicates. The authors propose Mg-C complexing on lines 116-117. Because Mg concentration is known, it was possible to check whether predicted solution composition matches that observed experimentally. It does not, predictions are too low. On lines 135-136 the authors suggest that this indicates a need for additional Mg-bearing complexes which may be carbon or silicate species (note inconsistency with line 116-117 where only C species are mentioned). It is not a particularly big step to propose such species and use the experiments to derive the Gibbs free energy. Is it reasonable? What would these species be? Why would we not have noticed them before? The nature of such experiments will always dictate that the only constraint on additional species is concentration observed vs. that predicted. This simple but very robust constraint should be followed to its logical conclusion.

2. Elevated XCO₂. Along the same lines, the experiments with quartz and with forsterite + enstatite show higher fCO₂ (XCO₂) than predicted for graphite saturation. The equilibria can be rearranged to demonstrate that this must mean slight changes in either fO₂ or fH₂O (note missing "f" for fugacity before H₂O in equation 4). No explanation for the former is provided. It seems highly unlikely to be the explanation in view of the absence of redox sensitive components in the minerals under investigation. It is suggested that fH₂O could be the culprit. In this case this value would need to decrease, and not by very much. Again, the argument is disengaged at exactly the moment when a little extra effort would take the work to the next level. The required decrease in fH₂O (or increase in fO₂) can be quantified (as indicated in the caption to fig. 3). What does this mean for the H₂O activity? Is it consistent with the SiO₂ concentration? It is known and can be used to see if a reasonable H₂O activity and/or activity coefficient results. Ultimately, there are

good models for SiO₂ and aH₂O in quartz saturated H₂O-CO₂ fluids (Walther and Orville, 1983; Newton and Manning, 2009), both are cited in the manuscript and both could be consulted to see if the required fH₂O shift is consistent with predictions from this previous work.

Bottom line, this version of the manuscript is close to being excellent and publishable. But I would not support publication in Nature Communications unless the explanations can be made robust along the lines I have suggested (or others if that is where the thinking leads...).

Reviewer #3 (Remarks to the Author)

Review of the revised manuscript by Tumiatti et al. titled "Silicate dissolution boosts the CO₂ concentrations in subduction fluids"

I appreciate the authors' effort to revise the manuscript to address the critique from the first round of review. The authors have made an attempt to now present their work better in the context of subduction zone carbon cycle. But I must say that even after the revision I am having hard time seeing the importance of the work especially for a platform such as Nature Communications and in the context of how major the findings are for global carbon cycle. To be more specific, here are some specific comments on the revised version of the manuscript and the authors' rebuttal –

1. Studying graphite dissolution in MS-COH system, i.e., in the presence of olivine and enstatite is puzzling. I noted this before and the authors' response was that they wanted to study how graphite dissolves along the top interface of the slab. But what is the source of graphite in the base of the mantle wedge? If it is organic carbon that transforms to graphite, it should be in sediment, so I can see the reason of studying S-COH system as a simplified model of graphite dissolution in subducting sediment. But there is no justification of studying graphite dissolution in hydrous fluid in the presence of olivine and enstatite. This is critical because this choice seems to have led the authors to conduct their experiments at conditions that are NOT directly relevant for subduction zones (see below).

2. The authors argue that the reason they did not perform their experiments at higher pressures is because they would end up stabilizing magnesite in the MS-COH system and that could complicate in the interpretation of their results because it would be difficult to know whether CO₂ is coming from graphite or carbonate dissolution. This is no reason of doing the experiments at 1 GPa, because this reasoning does not hold for the S-COH system – magnesite cannot form in the S-COH system and that system is much more relevant.

3. Experimental conditions (1 GPa, 800 C) – The chosen P-T condition of the experiments does NOT apply to any subduction zone worldwide. The slab surface temperature at 1 GPa varies from as low as <50 degree C to as high as 400 degree C (see any subduction zone thermal models; e.g., Syracuse et al., 2010) depending on which subduction zone (cold versus hot) is chosen. So if the goal is to constrain forearc flux of CO₂ through graphite dissolution, the experiments should have been conducted at 400 degree or less (at 1 GPa). The relative change of CO₂ dissolution owing to solutes is interesting but what matters the most is what range of absolute values are relevant. The application of the authors' data (generated at a temperature that is 2-16 times more than what is relevant at the pressure of 1 GPa) to update the CO₂ dissolution in fluid in subduction zone is thus hugely problematic. Actually almost surely wrong. It is NOT okay to say that the previously estimated flux should be raised by 10-30% where we have no idea what the effect of temperature on such estimates are.

4. There is still no clear explanation why dissolved SiO₂ should increase CO₂ dissolution from graphite.

From all of the above it seems that the application of the data obtained to really understand subduction zone carbon cycle is an afterthought. The authors seem to have been mostly interested in measuring the fluid composition from quench experiments. This report at best can serve as a proof of concept for that as an extension of the Tiraboschi et al., paper, which already outlines the

method of the fluid analysis. But there is no data or explanation here, which can be applied to revise our understanding of subduction zone carbon cycle.

Comments by the three reviewers have been extremely helpful as they provided thoughtful suggestions for undertaking revisions of the manuscript. This revised version contains **new experiments, implemented thermodynamic models, a reorganized manuscript and new, figures and tables in the main text and in Supplementary Information**. We believe that the new experimental data, which confirm and strengthen our conclusion concerning the unexpected role of silica in boosting the CO₂ content in graphite-saturated fluids, provide a much more convincing indication of the potential geologic implications of this new mechanism for the global carbon flux.

In particular, in order to strengthen our conclusions and to check the reproducibility of our data, we have carried out the following

a) Two new experiments in the systems COH and S-COH at higher pressure (3 GPa, 800°C) to test whether the enhanced XCO₂ occurs at conditions more representative of natural subduction zones as suggested by Rev. 3 (new data in Fig. 1, Fig. S3, Table S1, Table S2)

b) A new dissolution experiment in the system MS-COH in order to check the enhanced solubility of Mg silicates (sample CZ29 has replaced sample CZ26 in Table S3);

c) Explicit thermodynamic modelling of the enhanced solubility of Mg silicates, providing calculated model values of the solubility of forsterite + enstatite in water and forsterite + enstatite in COH fluid with the Deep Earth Water model¹⁰ (new Table S5);

d) Extension of the thermodynamic modelling of elevated XCO₂, quantifying and comparing the activities of water and silica, as suggested by Rev. 2.

The main conclusions we obtained from these new results are as follows:

- 1) The enhanced XCO₂ can be observed at 1 GPa also occurs at 3 GPa;
- 2) The enhanced solubility of Mg silicates in COH fluids has been confirmed;
- 3) The enhanced Mg- and Si-solubility is feasible assuming new Si-C and Mg-Si-C complexes, likely organometallic. Such complexes would not be expected under the more oxidized conditions reported in previous experimental studies of silicate-CO₂-H₂O systems.
- 4) The elevated XCO₂ can be attained by decreasing the activity of water by around 4 log units, a value that is 20-40 times higher compared with the activity of silica. Therefore, we suggested that only a small fraction (0.3-1.89 mol%) of the measured dissolved silica is influencing the water activity forming monomers and dimers. This model would suggest that new Si-C complexes are formed (organosilicon?), probably not significant in more oxidized, graphite-free systems investigated in previous studies.

Below, we describe our revisions in response to the comments of the referees:

Reviewer 1

One of my major criticisms about publication of the manuscript "Silicate dissolution boosts the CO₂ concentrations in subduction fluids" by Tumiati et al. was that the focus of the work, or at least the discussion/conclusion was too much on the geochemistry/technical results without setting them into a "global" context. Here I fully agree with reviewer #3, who also stated that point. The authors, however, decided to address this point by adding a small paragraph about the impacts on the global carbon cycle, without changing the overall character of the manuscript. Therefore, the text reads now a little better and if you want you can imagine what the major impact of that work on

global element cycles is, but I am still not convinced that a broad audience would get that point. For example, how significant in terms of the "natural" vs. anthropogenic CO₂ budget and the carbon cycle is a 30% increase in carbon transfer from the subducted plate? For geochemists/petrologists/geodynamicists this is of course of fundamental relevance, but I think that Nature Communications is aiming for a broader audience.

We agree with the reviewer that the manuscript is not focused on the quantification of the carbon flux. Nevertheless, the manuscript addresses processes that are active on a global scale. In order to better quantify the carbon recycling in light of the new mechanism presented here, some additional data are needed, which are unfortunately either not available or not sufficiently constrained at present. In particular, the average content of organic matter in subducted rocks is still poorly constrained and published works do not fully account for organic carbon in pelagic sediments and turbidites (see Kelemen & Manning, 2015). Moreover, the quantification of the CO₂ release would require a full equation of state characterization of the standard partial molal properties of Si-C and Mg-Si-C-complexes, which are currently not known, and therefore we must await the development of estimation schemes for refining the properties of such complexes. We are working on that, and likely we will be able to address this topic in the near future.

Reviewer 2

This manuscript describes an intriguing set of observations from experiments on quartz and forsterite+enstatite in the presence of graphite and COH fluid at high temperature and pressure. There are two novel insights. The first is that there is higher solubility of Mg silicates in the COH fluid than in pure H₂O, which is unexpected. The second is that the CO₂ content of the fluid is higher than it should be if the only consideration is graphite saturation at a known fH₂ and fO₂. The revised manuscript has been significantly modified to address my earlier complaint that it is necessary to provide/propose a mechanism for these observations. However, the explanations for the two surprises are in each case speculative as the manuscript stands now. Both could be made quantitative, and therefore better support the hypotheses. In my view, this step is necessary to make it publishable in Nature Communications over a regular geochemistry journal. Bottom line, this version of the manuscript is close to being excellent and publishable. But I would not support publication in Nature Communications unless the explanations can be made robust along the lines I have suggested (or others if that is where the thinking leads...).

1. Enhanced solubility of Mg silicates. The authors propose Mg-C complexing on lines 116-117. Because Mg concentration is known, it was possible to check whether predicted solution composition matches that observed experimentally. It does not, predictions are too low. On lines 135-136 the authors suggest that this indicates a need for additional Mg-bearing complexes which may be carbon or silicate species (note inconsistency with line 116-117 where only C species are mentioned). It is not a particularly big step to propose such species and use the experiments to derive the Gibbs free energy. Is it reasonable? What would these species be? Why would we not have noticed them before? The nature of such experiments will always dictate that the only constraint on additional species is concentration observed vs. that predicted. This simple but very robust constraint should be followed to its logical conclusion.

Following these suggestions, in the revised manuscript (lines 136 – 140, Table S5), we present a quantification of a Mg(OH)₂ complex to account for the solubility of Mg in the C-free, forsterite + enstatite + water (MS-H) system. With graphite present, in the system MS-COH, we use the Mg(OH)₂ complex and an additional Mg-Si-C complex to account for the high solubilities of Mg and Si (lines 140 – 160, Table S5). As an example, we chose an Mg-Si-C complex involving reduced C-species, e.g. Mg[OSi(OH₃)] [CH₃CH₂COO] (Table S5). This specific complex not only provides an explanation for the distinctive enhanced solubilities of Mg and Si measured in the

present study, but also it is predicted to be important only in relatively reducing systems. It should not be significant at all in the COH fluids in all previous studies of Mg-silicate solubilities or mineral-CO₂-H₂O systems that have focused on more oxidizing conditions without graphite. This explains why such a species has not been noticed before. Analogous calculations for the S-COH system suggest the possibility of a complex involving Si and reduced C-species.

2. Elevated XCO₂. Along the same lines, the experiments with quartz and with forsterite + enstatite show higher fCO₂ (XCO₂) than predicted for graphite saturation. The equilibria can be rearranged to demonstrate that this must mean slight changes in either fO₂ or fH₂O (note missing “f” for fugacity before H₂O in equation 4). No explanation for the former is provided. It seems highly unlikely to be the explanation in view of the absence of redox sensitive components in the minerals under investigation. It is suggested that fH₂O could be the culprit. In this case this value would need to decrease, and not by very much. Again, the argument is disengaged at exactly the moment when a little extra effort would take the work to the next level. The required decrease in fH₂O (or increase in fO₂) can be quantified (as indicated in the caption to fig. 3). What does this mean for the H₂O activity? Is it consistent with the SiO₂ concentration? It is known and can be used to see if a reasonable H₂O activity and/or activity coefficient results. Ultimately, there are good models for SiO₂ and aH₂O in quartz saturated H₂O-CO₂ fluids (Walther and Orville, 1983; Newton and Manning, 2009), both are cited in the manuscript and both could be consulted to see if the required fH₂O shift is consistent with predictions from this previous work.

We agree with the reviewer that in view of the absence of redox sensitive components in the minerals under investigation, even small variations of fO₂ are highly unlikely in our experimental system and therefore we suggest that a decrease in fH₂O is the culprit of the observed increase in fCO₂ in the MS-COH and S-COH systems. By using the H₂O fugacity coefficient from Zhang & Duan (1.579 at 1 GPa and 800° C), we are able to calculate the water decrease in terms of activity aH₂O, assuming that aH₂O = fH₂O/fH₂O⁰, where fH₂O⁰ is the fugacity of pure water at 1 GPa and 800° C (i.e., 15785.5). The required decrease in aH₂O with respect to the pure COH system is -4.17 log units in the experiments buffered with NNO and -4.14 log units in the experiments buffered with FMQ. Because only small differences in measured XCO₂ have been observed between S-COH and MS-COH model, we argue that dissolved silica monomers (Si(OH)₄) and dimers (Si₂O(OH)₆) are much more effective than Mg-Si-C complexes in decreasing water activity. The activity of total silica can be calculated on the basis of the measured SiO₂ molality in the experiments buffered with NNO and the measured XCO₂ of the corresponding COH fluid, log (aSiO₂) being equal to -2.59 in MS-COH and -2.90 in the S-COH systems. The activity of dissolved silica in the S-COH and MS-COH systems is therefore much higher (about 20 times in S-COH system; 40 times in MS-COH system) compared to the difference in aH₂O estimated in the S-COH and MS-COH systems versus the pure COH system. In order to meet the observed increase in fCO₂, we estimated that only 0.31 mol% (S-COH system; i.e., 0.006 mSiO₂/kgH₂O) and 1.89 mol% (MS-COH system; i.e., 0.004 mSiO₂/kgH₂O) of the measured dissolved silica are required, assuming that the decrease in water activity is solely related to the formation of hydrated silica monomers (Si(OH)₄) and dimers (Si₂O(OH)₆). These solubility data are almost identical to quartz solubility in graphite-free systems bearing very high XCO₂, H₂O-CO₂ fluids (Newton & Manning, 2009), strongly suggesting that new Si-C complexes are formed (organosilicon?), probably not significant in more oxidized, graphite-free systems.

Reviewer 3

Review of the revised manuscript by Tumiatti et al. titled “Silicate dissolution boosts the CO₂ concentrations in subduction fluids”

I appreciate the authors’ effort to revise the manuscript to address the critique from the first round of review. The authors have made an attempt to now present their work better in the context of

subduction zone carbon cycle. But I must say that even after the revision I am having hard time seeing the importance of the work especially for a platform such as Nature Communications and in the context of how major the findings are for global carbon cycle. It seems that the application of the data obtained to really understand subduction zone carbon cycle is an afterthought. The authors seem to have been mostly interested in measuring the fluid composition from quench experiments. This report at best can serve as a proof of concept for that as an extension of the Tiraboschi et al., paper, which already outlines the method of the fluid analysis. But there is no data or explanation here, which can be applied to revise our understanding of subduction zone carbon cycle.

We believe that the revised version of the manuscript now addresses most of the concerns of the reviewer. Concerning the applicability to quantify the global carbon flux and the importance of our data, please refer to the reply to Reviewer 1. We removed from the main text and the supplementary material the detailed description of the piercing technique, which has been published in the meantime in a technical paper (Tiraboschi et al., 2016). However, the major conclusion of our paper, i.e., the unpredicted enhanced CO₂ of subduction fluids interacting with carbonaceous matter and silicates, were not addressed at all in the Tiraboschi et al. (2016) paper.

1. Studying graphite dissolution in MS-COH system, i.e., in the presence of olivine and enstatite is puzzling. I noted this before and the authors' response was that they wanted to study how graphite dissolves along the top interface of the slab. But what is the source of graphite in the base of the mantle wedge? If it is organic carbon that transforms to graphite, it should be in sediment, so I can see the reason of studying S-COH system as a simplified model of graphite dissolution in subducting sediment. But there is no justification of studying graphite dissolution in hydrous fluid in the presence of olivine and enstatite. This is critical because this choice seems to have led the authors to conduct their experiments at conditions that are NOT directly relevant for subduction zones (see below).

Because our model experiments aim to investigate fluids interacting with a subduction mélange, we believe that the interaction with mantle-derived rocks is interesting. We agree that graphite should occur in metasediments, so we can easily imagine that graphite-saturated COH fluids will interact both with slices of mantle-rocks in the mélange and the overlying mantle wedge, too. Therefore, we need the constraints provided by experiments in the MS-COH system.

2. The authors argue that the reason they did not perform their experiments at higher pressures is because they would end up stabilizing magnesite in the MS-COH system and that could complicate in the interpretation of their results because it would be difficult to know whether CO₂ is coming from graphite or carbonate dissolution. This is no reason of doing the experiments at 1 GPa, because this reasoning does not hold for the S-COH system – magnesite cannot form in the S-COH system and that system is much more relevant.

We added two experiments at much higher pressure, i.e. 3 GPa and 800°C, one performed in the simple COH system and one in the S-COH system (e.g. lines 81 – 86, and the following discussion). At 3 GPa we observed the same effect of silicates in enhancing the CO₂ content of the fluid as previously documented for experiments at 1 GPa. Experiments at 3 GPa and 800°C in the MS-COH system involving magnesite will be the topic of future work.

3. Experimental conditions (1 GPa, 800 C) – The chosen P-T condition of the experiments does NOT apply to any subduction zone worldwide. The slab surface temperature at 1 GPa varies from as low as <50 degree C to as high as 400 degree C (see any subduction zone thermal models; e.g., Syracuse et al., 2010) depending on which subduction zone (cold versus hot) is chosen. So if the

goal is to constrain forearc flux of CO₂ through graphite dissolution, the experiments should have been conducted at 400 degree or less (at 1 GPa). The relative change of CO₂ dissolution owing to solutes is interesting but what matters the most is what range of absolute values are relevant. The application of the authors' data (generated at a temperature that is 2-16 times more than what is relevant at the pressure of 1 GPa) to update the CO₂ dissolution in fluid in subduction zone is thus hugely problematic. Actually almost surely wrong. It is NOT okay to say that the previously estimated flux should be raised by 10-30% where we have no idea what the effect of temperature on such estimates are.

Our new experiments at 3 GPa extend our results into a broad range of subduction-zone conditions. We emphasize in the revised text (lines 239-259) that the CO₂-content of fluids in subduction zones could be about 30% higher than previously predicted because of the new enhancement mechanism indicated by our experimental data.

We agree with the reviewer that even more experimental conditions over a wider range of pressures and temperatures are needed to truly test the potential impact on the global carbon flux. We have emphasized this in the revised text on lines 257 – 259.

4. There is still no clear explanation why dissolved SiO₂ should increase CO₂ dissolution from graphite.

Fixed. Please refer to reply 2 to Reviewer 2.

Reviewers' Comments:

Reviewer #2:

Remarks to the Author:

This version of the manuscript very nicely addresses the two major issues I raised in the last round of reviews. I believe the paper is now much stronger, because it provides clear hypotheses - supported by data and calculation - to explain the unexpected experimental results. In particular, the analysis of the elevated solubility has led to the proposal of a new complex involving Si, Mg and reduced C. The tests are taken to the right level and supported, even if provisionally, by the experimental data.

If it is correct that the proposed complexing occurs, we have learned something new and fundamental about COH fluids in the presence of silica. I strongly support publication of this current version of the paper.

Reviewer #3:

Remarks to the Author:

The revised manuscript by Tumiatti et al. now includes few more experiments at higher pressures and temperatures that are more relevant for the subduction zones. The authors find similarly 'enhanced' solubility of CO₂ at higher pressures as in their low-P experiments. The expanded data set definitely are more valuable than before. However, the explanation of the authors' observations and the implications of the new 'evidence' for enhanced dissolution of CO₂ in fluid remain poor. First of all, the authors now suggest that their observed enhanced dissolution must call for Si-C and Mg-Si-C complexes in the fluid because their systems are 'reduced'. But the presence of graphite does not necessarily make the system reduced. Having both graphite and a CO₂-rich fluid sets the oxygen fugacity of the system very close to CCO buffer, which is not different from most shallow upper mantle conditions where magma or fluids are generated. There has been no report or evidence of Si-C or Mg-Si-C complexes either in fluid or melt at these types of conditions. In fact, experimental work in far more reducing conditions than this study also never found Si-C complexes. So to suggest that these extremely reduced complexes exist in fluid and that is the explanation for their enhanced CO₂ dissolution is simply unfounded. Without any spectroscopic evidence of such, currently this suggestion would not hold much weight.

30% increase in the dissolution of carbon in subduction zone fluid is also not a significant enough increase that this has much importance for global carbon fluxes in and out of the subduction zones.

The paper is also surprisingly not up-to-date with many papers that in the past have thoroughly dealt with subduction efficiency of carbon. Some that come to mind, which I would expect to be acknowledged and discussed are - Galvez et al. (2016 - Nature), Dasgupta (2013 - RiMG), Hirschmann and Dasgupta (2009 - ChemGeol).

Reviewer 2

This version of the manuscript very nicely addresses the two major issues I raised in the last round of reviews. I believe the paper is now much stronger, because it provides clear hypotheses - supported by data and calculation - to explain the unexpected experimental results. In particular, the analysis of the elevated solubility has led to the proposal of a new complex involving Si, Mg and reduced C. The tests are taken to the right level and supported, even if provisionally, by the experimental data. If it is correct that the proposed complexing occurs, we have learned something new and fundamental about COH fluids in the presence of silica. I strongly support publication of this current version of the paper.

We thank Reviewer #2 for the constructive past reviews. We are glad that he now supports publication of our paper in Nature Communications

Reviewer 3

i) The revised manuscript by Tumiatti et al. now includes few more experiments at higher pressures and temperatures that are more relevant for the subduction zones. The authors find similarly 'enhanced' solubility of CO₂ at higher pressures as in their low-P experiments. The expanded data set definitely are more valuable than before.

We are glad that the reviewer now recognizes the value of our experiments and database. He forced us to run experiments at conditions more relevant for subduction zones and we believe that the manuscript benefited from them.

ii) However, the explanation of the authors' observations and the implications of the new 'evidence' for enhanced dissolution of CO₂ in fluid remain poor. First of all, the authors now suggest that their observed enhanced dissolution must call for Si-C and Mg-Si-C complexes in the fluid because their systems are 'reduced'. But the presence of graphite does not necessarily make the system reduced. Having both graphite and a CO₂-rich fluid sets the oxygen fugacity of the system very close to CCO buffer, which is not different from most shallow upper mantle conditions where magma or fluids are generated. There has been no report or evidence of Si-C or Mg-Si-C complexes either in fluid or melt at these types of conditions. In fact, experimental work in far more reducing conditions than this study also never found Si-C complexes.

We agree that our experiments are close to the CCO buffer, and their redox state approaches that of the slab-mantle interface ($\Delta\text{FMQ} = -0.47$ to -0.70). Nevertheless, at these conditions the stability of organic compounds is not impossible. We believe that a major misunderstanding has occurred here. The reviewer is very concerned about the suggested presence of Si-C and Mg-Si-C complexes, and he states that these compounds are "extremely reduced". They are not. We meant compounds containing Si-O-C and Si-O-Mg bonds. Perhaps the reviewer has mistaken our proposed species for silicon carbide species, i.e. species containing Si-C bonds, which we agree might only be stable at ultra-reducing conditions (e.g., Schmidt et al., 2014, Progress in Earth and Planetary Science).

In order to eliminate such misunderstandings, we have modified the main text in order to make some concepts clearer and less prone to misunderstanding.

iii) So to suggest that these extremely reduced complexes exist in fluid and that is the explanation for their enhanced CO₂ dissolution is simply unfounded. Without any spectroscopic evidence of such, currently this suggestion would not hold much weight.

Other papers suggested the presence of organic compounds in subduction fluids (e.g. Sverjensky et al. 2014, Nature Geoscience), without providing any spectroscopic evidence. We are performing new experiments, aiming at measuring organic compounds in COH fluids, but we expect to obtain results within the next 2 years or so. In any case, the main point of our experimental study is the remarkable increase in CO₂ concentrations associated with the addition of silicates to the graphite+water system. Thanks to the review system we have now demonstrated the potential importance of this effect over a wide range of pressures and temperatures relevant to subduction zone systems.

iv) 30% increase in the dissolution of carbon in subduction zone fluid is also not a significant enough increase that this has much importance for global carbon fluxes in and out of the subduction zones.

This paper focuses on a really novel process, which challenge the thermodynamic modelling of COH fluids, and in particular its use in order to retrieve the CO₂ content of subduction fluids. Although the aim of the paper is not the quantification of carbon released in subduction zone, which must await other experimental data, we clearly state in our manuscript that "Other experiments for more complex carbonate-bearing systems and a better quantification of the content of graphite and organic matter in subducted sediments are needed in order to better quantify the impact of this novel mechanism on the global carbon flux".

v) The paper is also surprisingly not up-to-date with many papers that in the past have thoroughly dealt with subduction efficiency of carbon. Some that come to mind, which I would expect to be acknowledged and discussed are - Galvez et al. (2016 - Nature), Dasgupta (2013 - RiMG), Hirschmann and Dasgupta (2009 - ChemGeol).

Fixed. We added also other recent pertinent references.

Reviewers' Comments:

Reviewer #2:

None